# Supervised Policy Update for Deep Reinforcement Learning

**Quan Vuong**
University of California, San Diego
qvuong@ucsd.edu

**Yiming Zhang**
New York University
yiming.zhang@cs.nyu.edu

**Keith Ross**
New York University/New York University Shanghai
keithwross@nyu.edu

## Abstract

We propose a new sample-efficient methodology, called Supervised Policy Update (SPU), for deep reinforcement learning. Starting with data generated by the current policy, SPU formulates and solves a constrained optimization problem in the non-parameterized proximal policy space. Using supervised regression, it then converts the optimal non-parameterized policy to a parameterized policy, from which it draws new samples. The methodology is general in that it applies to both discrete and continuous action spaces, and can handle a wide variety of proximity constraints for the non-parameterized optimization problem. We show how the Natural Policy Gradient and Trust Region Policy Optimization (NPG/TRPO) problems, and the Proximal Policy Optimization (PPO) problem can be addressed by this methodology. The SPU implementation is much simpler than TRPO. In terms of sample efficiency, our extensive experiments show SPU outperforms TRPO in Mujoco simulated robotic tasks and outperforms PPO in Atari video game tasks.

## 1 Introduction

The policy gradient problem in deep reinforcement learning (DRL) can be defined as seeking a parameterized policy with high expected reward. An issue with policy gradient methods is poor sample efficiency (Kakade, 2003; Schulman et al., 2015a; Wang et al., 2016b; Wu et al., 2017; Schulman et al., 2017). In algorithms such as REINFORCE (Williams, 1992), new samples are needed for every gradient step. When generating samples is expensive (such as robotic environments), sample efficiency is of central concern. The sample efficiency of an algorithm is defined to be the number of calls to the environment required to attain a specified performance level (Kakade, 2003).

Thus, given the current policy and a fixed number of trajectories (samples) generated, the goal of the sample efficiency problem is to construct a new policy with the highest performance improvement possible. To do so, it is desirable to limit the search to policies that are close to the original policy $\pi_{\theta_k}$ (Kakade, 2002; Schulman et al., 2015a; Wu et al., 2017; Achiam et al., 2017; Schulman et al., 2017; Tangkaratt et al., 2018). Intuitively, if the candidate new policy $\pi_\theta$ is far from the original policy $\pi_{\theta_k}$, it may not perform better than the original policy because too much emphasis is being placed on the relatively small batch of new data generated by $\pi_{\theta_k}$, and not enough emphasis is being placed on the relatively large amount of data and effort previously used to construct $\pi_{\theta_k}$.

This guideline of limiting the search to nearby policies seems reasonable in principle, but requires a distance $\eta(\pi_\theta, \pi_{\theta_k})$ between the current policy $\pi_{\theta_k}$ and the candidate new policy $\pi_\theta$, and then attempt to solve the constrained optimization problem:

$$\underset{\theta}{\text{maximize}} \quad \hat{J}(\pi_\theta \mid \pi_{\theta_k}, \text{new data}) \tag{1}$$

$$\text{subject to} \quad \eta(\pi_\theta, \pi_{\theta_k}) \leq \delta \tag{2}$$

where $\hat{J}(\pi_\theta \mid \pi_{\theta_k}, \text{new data})$ is an estimate of $J(\pi_\theta)$, the performance of policy $\pi_\theta$, based on the previous policy $\pi_{\theta_k}$ and the batch of fresh data generated by $\pi_{\theta_k}$. The objective (1) attempts to

maximize the performance of the updated policy, and the constraint (2) ensures that the updated policy is not too far from the policy $\pi_{\theta_k}$ that was used to generate the data. Several recent papers (Kakade, 2002; Schulman et al., 2015a; 2017; Tangkaratt et al., 2018) belong to the framework (1)-(2).

Our work also strikes the right balance between performance and simplicity. The implementation is only slightly more involved than PPO (Schulman et al., 2017). Simplicity in RL algorithms has its own merits. This is especially useful when RL algorithms are used to solve problems outside of traditional RL testbeds, which is becoming a trend (Zoph & Le, 2016; Mingxing Tan, 2018).

We propose a new methodology, called Supervised Policy Update (SPU), for this sample efficiency problem. The methodology is general in that it applies to both discrete and continuous action spaces, and can address a wide variety of constraint types for (2). Starting with data generated by the current policy, SPU optimizes over a proximal policy space to find an optimal *non-parameterized policy*. It then solves a supervised regression problem to convert the non-parameterized policy to a parameterized policy, from which it draws new samples. We develop a general methodology for finding an optimal policy in the non-parameterized policy space, and then illustrate the methodology for three different definitions of proximity. We also show how the Natural Policy Gradient and Trust Region Policy Optimization (NPG/TRPO) problems and the Proximal Policy Optimization (PPO) problem can be addressed by this methodology. While SPU is substantially simpler than NPG/TRPO in terms of mathematics and implementation, our extensive experiments show that SPU is more sample efficient than TRPO in Mujoco simulated robotic tasks and PPO in Atari video game tasks.

Off-policy RL algorithms generally achieve better sample efficiency than on-policy algorithms (Haarnoja et al., 2018). However, the performance of an on-policy algorithm can usually be substantially improved by incorporating off-policy training (Mnih et al. (2015), Wang et al. (2016a)). Our paper focuses on igniting interests in separating finding the optimal policy into a two-step process: finding the optimal non-parameterized policy, and then parameterizing this optimal policy. We also wanted to deeply understand the on-policy case before adding off-policy training. We thus compare with algorithms operating under the same algorithmic constraints, one of which is being on-policy. We leave the extension to off-policy to future work. We do not claim state-of-the-art results.

## 2 PRELIMINARIES

We consider a Markov Decision Process (MDP) with state space $\mathcal{S}$, action space $\mathcal{A}$, and reward function $r(s, a)$, $s \in \mathcal{S}$, $a \in \mathcal{A}$. Let $\pi = \{\pi(a|s) : s \in \mathcal{S}, a \in \mathcal{A}\}$ denote a policy, let $\Pi$ be the set of all policies, and let the expected discounted reward be:

$$J(\pi) \triangleq \mathop{\mathbb{E}}_{\tau \sim \pi} \left[ \sum_{t=0}^{\infty} \gamma^t r(s_t, a_t) \right] \tag{3}$$

where $\gamma \in (0, 1)$ is a discount factor and $\tau = (s_0, a_0, s_1, \dots)$ is a sample trajectory. Let $A^\pi(s, a)$ be the advantage function for policy $\pi$ (Levine, 2017). Deep reinforcement learning considers a set of parameterized policies $\Pi_{\text{DL}} = \{\pi_\theta | \theta \in \Theta\} \subset \Pi$, where each policy is parameterized by a neural network called the policy network. In this paper, we will consider optimizing over the parameterized policies in $\Pi_{\text{DL}}$ as well as over the non-parameterized policies in $\Pi$. For concreteness, we assume that the state and action spaces are finite. However, our methodology also applies to continuous state and action spaces, as shown in the Appendix.

One popular approach to maximizing $J(\pi_\theta)$ over $\Pi_{\text{DL}}$ is to apply stochastic gradient ascent. The gradient of $J(\pi_\theta)$ evaluated at a specific $\theta = \theta_k$ can be shown to be (Williams, 1992):

$$\nabla_\theta J(\pi_{\theta_k}) = \mathop{\mathbb{E}}_{\tau \sim \pi_{\theta_k}} \left[ \sum_{t=0}^{\infty} \gamma^t \nabla_\theta \log \pi_{\theta_k}(a_t|s_t) A^{\pi_{\theta_k}}(s_t, a_t) \right]. \tag{4}$$

We can approximate (4) by sampling N trajectories of length T from $\pi_{\theta_k}$:

$$\nabla_\theta J(\pi_{\theta_k}) \approx \frac{1}{N} \sum_{i=1}^{N} \sum_{t=0}^{T-1} \nabla_\theta \log \pi_{\theta_k}(a_{it}|s_{it}) A^{\pi_{\theta_k}}(s_t, a_t) \triangleq g_k \tag{5}$$

Additionally, define $d^\pi(s) \triangleq (1 - \gamma) \sum_{t=0}^{\infty} \gamma^t P_\pi(s_t = s)$ for the the future state probability distribution for policy $\pi$, and denote $\pi(\cdot|s)$ for the probability distribution over the action space $\mathcal{A}$

when in state $s$ and using policy $\pi$. Further denote $D_{\mathrm{KL}}(\pi \parallel \pi_{\theta_k})[s]$ for the KL divergence from $\pi(\cdot|s)$ to $\pi_{\theta_k}(\cdot|s)$, and denote the following as the "aggregated KL divergence".

$$\bar{D}_{\mathrm{KL}}(\pi \parallel \pi_{\theta_k}) = \mathop{\mathbb{E}}_{s \sim d^{\pi_{\theta_k}}} [D_{\mathrm{KL}}(\pi \parallel \pi_{\theta_k})[s]] \tag{6}$$

## 2.1 Surrogate Objectives for the Sample Efficiency Problem

For the sample efficiency problem, the objective $J(\pi_\theta)$ is typically approximated using samples generated from $\pi_{\theta_k}$ (Schulman et al., 2015a; Achiam et al., 2017; Schulman et al., 2017). Two different approaches are typically used to approximate $J(\pi_\theta) - J(\pi_{\theta_k})$. We can make a first order approximation of $J(\pi_\theta)$ around $\theta_k$ (Kakade, 2002; Peters & Schaal, 2008a;b; Schulman et al., 2015a):

$$J(\pi_\theta) - J(\pi_{\theta_k}) \approx (\theta - \theta_k)^T \nabla_\theta J(\pi_{\theta_k}) \approx (\theta - \theta_k)^T g_k \tag{7}$$

where $g_k$ is the sample estimate (5). The second approach is to approximate the state distribution $d^\pi(s)$ with $d^{\pi_{\theta_k}}(s)$ (Achiam et al., 2017; Schulman et al., 2017; Achiam, 2017):

$$J(\pi) - J(\pi_{\theta_k}) \approx \mathcal{L}_{\pi_{\theta_k}}(\pi) \triangleq \frac{1}{1-\gamma} \mathop{\mathbb{E}}_{s \sim d^{\pi_{\theta_k}}} \mathop{\mathbb{E}}_{a \sim \pi_{\theta_k}(\cdot|s)} \left[ \frac{\pi(a|s)}{\pi_{\theta_k}(a|s)} A^{\pi_{\theta_k}}(s, a) \right] \tag{8}$$

There is a well-known bound for the approximation (8) (Kakade & Langford, 2002; Achiam et al., 2017). Furthermore, the approximation $\mathcal{L}_{\pi_{\theta_k}}(\pi_\theta)$ matches $J(\pi_\theta) - J(\pi_{\theta_k})$ to the first order with respect to the parameter $\theta$ (Achiam et al., 2017).

## 3 Related Work

Natural gradient (Amari, 1998) was first introduced to policy gradient by Kakade (Kakade, 2002) and then in (Peters & Schaal, 2008a;b; Achiam, 2017; Schulman et al., 2015a). referred to collectively here as NPG/TRPO. Algorithmically, NPG/TRPO finds the gradient update by solving the sample efficiency problem (1)-(2) with $\eta(\pi_\theta, \pi_{\theta_k}) = \bar{D}_{\mathrm{KL}}(\pi_\theta \parallel \pi_{\theta_k})$, i.e., use the aggregate KL-divergence for the policy proximity constraint (2). NPG/TRPO addresses this problem in the parameter space $\theta \in \Theta$. First, it approximates $J(\pi_\theta)$ with the first-order approximation (7) and $\bar{D}_{\mathrm{KL}}(\pi_\theta \parallel \pi_{\theta_k})$ using a similar second-order method. Second, it uses samples from $\pi_{\theta_k}$ to form estimates of these two approximations. Third, using these estimates (which are functions of $\theta$), it solves for the optimal $\theta^*$. The optimal $\theta^*$ is a function of $g_k$ and of $h_k$, the sample average of the Hessian evaluated at $\theta_k$. TRPO also limits the magnitude of the update to ensure $\bar{D}_{\mathrm{KL}}(\pi_\theta \parallel \pi_{\theta_k}) \leq \delta$ (i.e., ensuring the sampled estimate of the aggregated KL constraint is met *without* the second-order approximation).

SPU takes a very different approach by first (i) posing and solving the optimization problem in the non-parameterized policy space, and then (ii) solving a supervised regression problem to find a parameterized policy that is near the optimal non-parameterized policy. A recent paper, Guided Actor Critic (GAC), independently proposed a similar decomposition (Tangkaratt et al., 2018). However, GAC is much more restricted in that it considers only one specific constraint criterion (aggregated reverse-KL divergence) and applies only to continuous action spaces. Furthermore, GAC incurs significantly higher computational complexity, e.g. at every update, it minimizes the dual function to obtain the dual variables using SLSQP. MPO also independently propose a similar decomposition (Abbas Abdolmaleki, 2018). MPO uses much more complex machinery, namely, Expectation Maximization to address the DRL problem. However, MPO has only demonstrates preliminary results on problems with discrete actions whereas our approach naturally applies to problems with either discrete or continuous actions. In both GAC and MPO, working in the non-parameterized space is a by-product of applying the main ideas in those papers to DRL. Our paper demonstrates that the decomposition alone is a general and useful technique for solving constrained policy optimization.

Clipped-PPO (Schulman et al., 2017) takes a very different approach to TRPO. At each iteration, PPO makes many gradient steps while only using the data from $\pi_{\theta_k}$. Without the clipping, PPO is the approximation (8). The clipping is analogous to the constraint (2) in that it has the goal of keeping $\pi_\theta$ close to $\pi_{\theta_k}$. Indeed, the clipping keeps $\pi_\theta(a_t|s_t)$ from becoming neither much larger than $(1 + \epsilon)\pi_{\theta_k}(a_t|s_t)$ nor much smaller than $(1 - \epsilon)\pi_{\theta_k}(a_t|s_t)$. Thus, although the clipped PPO objective does not squarely fit into the optimization framework (1)-(2), it is quite similar in spirit. We note that the PPO paper considers adding the KL penalty to the objective function, whose gradient

is similar to ours. However, this form of gradient was demonstrated to be inferior to Clipped-PPO. To the best of our knowledge, it is only until our work that such form of gradient is demonstrated to outperform Clipped-PPO.

Actor-Critic using Kronecker-Factored Trust Region (ACKTR) (Wu et al., 2017) proposed using Kronecker-factored approximation curvature (K-FAC) to update both the policy gradient and critic terms, giving a more computationally efficient method of calculating the natural gradients. ACER (Wang et al., 2016a) exploits past episodes, linearizes the KL divergence constraint, and maintains an average policy network to enforce the KL divergence constraint. In future work, it would of interest to extend the SPU methodology to handle past episodes. In contrast to bounding the KL divergence on the action distribution as we have done in this work, Relative Entropy Policy Search considers bounding the joint distribution of state and action and was only demonstrated to work for small problems (Jan Peters, 2010).

## 4  SPU FRAMEWORK

The SPU methodology has two steps. In the first step, for a given constraint criterion $\eta(\pi, \pi_{\theta_k}) \leq \delta$, we find the optimal solution to the non-parameterized problem:

$$\underset{\pi \in \Pi}{\text{maximize}} \quad \mathcal{L}_{\pi_{\theta_k}}(\pi) \tag{9}$$

$$\text{subject to} \quad \eta(\pi, \pi_{\theta_k}) \leq \delta \tag{10}$$

Note that $\pi$ is not restricted to the set of parameterized policies $\Pi_{DL}$. As commonly done, we approximate the objective function (8). However, unlike PPO/TRPO, we are not approximating the constraint (2). We will show below the optimal solution $\pi^*$ for the non-parameterized problem (9)-(10) can be determined nearly in closed form for many natural constraint criteria $\eta(\pi, \pi_{\theta_k}) \leq \delta$.

In the second step, we attempt to find a policy $\pi_\theta$ in the parameterized space $\Pi_{DL}$ that is close to the target policy $\pi^*$. Concretely, to advance from $\theta_k$ to $\theta_{k+1}$, we perform the following steps:

 (i) We first sample $N$ trajectories using policy $\pi_{\theta_k}$, giving sample data $(s_i, a_i, A_i)$, $i = 1, .., m$. Here $A_i$ is an estimate of the advantage value $A^{\pi_{\theta_k}}(s_i, a_i)$. (For simplicity, we index the samples with $i$ rather than with $(i, t)$ corresponding to the $t$th sample in the $i$th trajectory.)

 (ii) For each $s_i$, we define the target distribution $\pi^*$ to be the optimal solution to the constrained optimization problem (9)-(10) for a specific constraint $\eta$.

(iii) We then fit the policy network $\pi_\theta$ to the target distributions $\pi^*(\cdot|s_i)$, $i = 1, .., m$. Specifically, to find $\theta_{k+1}$, we minimize the following supervised loss function:

$$L(\theta) = \frac{1}{N} \sum_{i=1}^{m} D_{\text{KL}}(\pi_\theta \parallel \pi^*)[s_i] \tag{11}$$

For this step, we initialize with the weights for $\pi_{\theta_k}$. We minimize the loss function $L(\theta)$ with stochastic gradient descent methods. The resulting $\theta$ becomes our $\theta_{k+1}$.

## 5  SPU APPLIED TO SPECIFIC PROXIMITY CRITERIA

To illustrate the SPU methodology, for three different but natural types of proximity constraints, we solve the corresponding non-parameterized optimization problem and derive the resulting gradient for the SPU supervised learning problem. We also demonstrate that different constraints lead to very different but intuitive forms of the gradient update.

### 5.1 FORWARD AGGREGATE AND DISAGGREGATE KL CONSTRAINTS

We first consider constraint criteria of the form:

$$\underset{\pi \in \Pi}{\text{maximize}} \quad \sum_s d^{\pi_{\theta_k}}(s) \underset{a \sim \pi(\cdot|s)}{\mathbb{E}} [A^{\pi_{\theta_k}}(s,a)] \tag{12}$$

$$\text{subject to} \quad \sum_s d^{\pi_{\theta_k}}(s) D_{\text{KL}}(\pi \parallel \pi_{\theta_k})[s] \leq \delta \tag{13}$$

$$D_{\text{KL}}(\pi \parallel \pi_{\theta_k})[s] \leq \epsilon \text{ for all } s \tag{14}$$

Note that this problem is equivalent to minimizing $\mathcal{L}_{\pi_{\theta_k}}(\pi)$ subject to the constraints (13) and (14). We refer to (13) as the "aggregated KL constraint" and to (14) as the "disaggregated KL constraint". These two constraints taken together restrict $\pi$ from deviating too much from $\pi_{\theta_k}$. We shall refer to (12)-(14) as the forward-KL non-parameterized optimization problem.

Note that this problem without the disaggregated constraints is analogous to the TRPO problem. The TRPO paper actually prefers enforcing the disaggregated constraint to enforcing the aggregated constraints. However, for mathematical conveniences, they worked with the aggregated constraints: "While it is motivated by the theory, this problem is impractical to solve due to the large number of constraints. Instead, we can use a heuristic approximation which considers the average KL divergence" (Schulman et al., 2015a). The SPU framework allows us to solve the optimization problem with the disaggregated constraints exactly. Experimentally, we compared against TRPO in a controlled experimental setting, e.g. using the same advantage estimation scheme, etc. Since we clearly outperform TRPO, we argue that SPU's two-process procedure has significant potentials.

For each $\lambda > 0$, define: $\pi^\lambda(a|s) = \frac{\pi_{\theta_k}(a|s)}{Z_\lambda(s)} e^{A^{\pi_{\theta_k}}(s,a)/\lambda}$ where $Z_\lambda(s)$ is the normalization term. Note that $\pi^\lambda(a|s)$ is a function of $\lambda$. Further, for each s, let $\lambda_s$ be such that $D_{\text{KL}}(\pi^{\lambda_s} \parallel \pi_{\theta_k})[s] = \epsilon$. Also let $\Gamma^\lambda = \{s : D_{\text{KL}}(\pi^\lambda \parallel \pi_{\theta_k})[s] \leq \epsilon\}$.

**Theorem 1** *The optimal solution to the problem* (12)-(14) *is given by:*

$$\tilde{\pi}^\lambda(a|s) = \begin{cases} \pi^\lambda(a|s) & s \in \Gamma^\lambda \\ \pi^{\lambda_s}(a|s) & s \notin \Gamma^\lambda \end{cases} \tag{15}$$

*where $\lambda$ is chosen so that $\sum_s d^{\pi_{\theta_k}}(s) D_{KL}(\tilde{\pi}^\lambda \parallel \pi_{\theta_k})[s] = \delta$ (Proof in subsection A.1).*

Equation (15) provides the structure of the optimal non-parameterized policy. As part of the SPU framework, we then seek a parameterized policy $\pi_\theta$ that is close to $\tilde{\pi}^\lambda(a|s)$, that is, minimizes the loss function (11). For each sampled state $s_i$, a straightforward calculation shows (Appendix B):

$$\nabla_\theta D_{\text{KL}}(\pi_\theta \parallel \tilde{\pi}^\lambda)[s_i] = \nabla_\theta D_{\text{KL}}(\pi_\theta \parallel \pi_{\theta_k})[s_i] - \frac{1}{\tilde{\lambda}_{s_i}} \underset{a \sim \pi_{\theta_k}(\cdot|s_i)}{\mathbb{E}} [\nabla_\theta \log \pi_\theta(a|s_i) A^{\pi_{\theta_k}}(s_i,a)] \tag{16}$$

where $\tilde{\lambda}_{s_i} = \lambda$ for $s_i \in \Gamma^\lambda$ and $\tilde{\lambda}_{s_i} = \lambda_{s_i}$ for $s_i \notin \Gamma^\lambda$. We estimate the expectation in (16) with the sampled action $a_i$ and approximate $A^{\pi_{\theta_k}}(s_i,a_i)$ as $A_i$ (obtained from the critic network), giving:

$$\nabla_\theta D_{\text{KL}}(\pi_\theta \parallel \tilde{\pi}^\lambda)[s_i] \approx \nabla_\theta D_{\text{KL}}(\pi_\theta \parallel \pi_{\theta_k})[s_i] - \frac{1}{\tilde{\lambda}_{s_i}} \frac{\nabla_\theta \pi_\theta(a_i|s_i)}{\pi_{\theta_k}(a_i|s_i)} A_i \tag{17}$$

To simplify the algorithm, we slightly modify (17). We replace the hyper-parameter $\delta$ with the hyper-parameter $\lambda$ and tune $\lambda$ rather than $\delta$. Further, we set $\tilde{\lambda}_{s_i} = \lambda$ for all $s_i$ in (17) and introduce per-state acceptance to enforce the disaggregated constraints, giving the approximate gradient:

$$\nabla_\theta D_{\text{KL}}(\pi_\theta \parallel \tilde{\pi}^\lambda) \approx \frac{1}{m} \sum_{i=1}^m [\nabla_\theta D_{\text{KL}}(\pi_\theta \parallel \pi_{\theta_k})[s_i] - \frac{1}{\lambda} \frac{\nabla_\theta \pi_\theta(a_i|s_i)}{\pi_{\theta_k}(a_i|s_i)} A_i] \mathbb{1}_{D_{\text{KL}}(\pi_\theta \parallel \pi_{\theta_k})[s_i] \leq \epsilon} \tag{18}$$

We make the approximation that the disaggregated constraints are only enforced on the states in the sampled trajectories. We use (18) as our gradient for supervised training of the policy

network. The equation (18) has an intuitive interpretation: the gradient represents a trade-off between the approximate performance of $\pi_\theta$ (as captured by $\frac{1}{\lambda} \frac{\nabla_\theta \pi_\theta(a_i|s_i)}{\pi_{\theta_k}(a_i|s_i)} A_i$) and how far $\pi_\theta$ diverges from $\pi_{\theta_k}$ (as captured by $\nabla_\theta D_{\text{KL}}(\pi_\theta \parallel \pi_{\theta_k})[s_i]$). For the stopping criterion, we train until $\frac{1}{m} \sum_i D_{\text{KL}}(\pi_\theta \parallel \pi_{\theta_k})[s_i] \approx \delta$.

## 5.2  BACKWARD KL CONSTRAINT

In a similar manner, we can derive the structure of the optimal policy when using the reverse KL-divergence as the constraint. For simplicity, we provide the result for when there are only disaggregated constraints. We seek to find the non-parameterized optimal policy by solving:

$$\underset{\pi \in \Pi}{\text{maximize}} \quad \sum_s d^{\pi_{\theta_k}}(s) \underset{a \sim \pi(\cdot|s)}{\mathbb{E}} \left[ A^{\pi_{\theta_k}}(s, a) \right] \tag{19}$$

$$D_{\text{KL}}(\pi \parallel \pi_{\theta_k})[s] \leq \epsilon \text{ for all } s \tag{20}$$

**Theorem 2** *The optimal solution to the problem* (19)-(20) *is given by:*

$$\pi^*(a|s) = \pi_{\theta_k}(a|s) \frac{\lambda(s)}{\lambda'(s) - A^{\pi_{\theta_k}}(s, a)} \tag{21}$$

*where* $\lambda(s) > 0$ *and* $\lambda'(s) > \max_a A^{\pi_{\theta_k}}(s, a)$ *(Proof in subsection A.2).*

Note that the structure of the optimal policy with the backward KL constraint is quite different from that with the forward KL constraint. A straight forward calculation shows (Appendix B):

$$\nabla_\theta D_{\text{KL}}(\pi_\theta \parallel \pi^*)[s] = \nabla_\theta D_{\text{KL}}(\pi_\theta \parallel \pi_{\theta_k})[s] - \underset{a \sim \pi_{\theta_k}}{E} \left[ \frac{\nabla_\theta \pi_\theta(a|s)}{\pi_{\theta_k}(a|s)} \log \left( \frac{1}{\lambda'(s) - A^{\pi_{\theta_k}}(s, a)} \right) \right] \tag{22}$$

The equation (22) has an intuitive interpretation. It increases the probability of action $a$ if $A^{\pi_{\theta_k}}(s, a) > \lambda'(s) - 1$ and decreases the probability of action $a$ if $A^{\pi_{\theta_k}}(s, a) < \lambda'(s) - 1$. (22) also tries to keep $\pi_\theta$ close to $\pi_{\theta_k}$ by minimizing their KL divergence.

## 5.3  $L^\infty$ CONSTRAINT

In this section we show how a PPO-like objective can be formulated in the context of SPU. Recall from Section 3 that the the clipping in PPO can be seen as an attempt at keeping $\pi_\theta(a_i|s_i)$ from becoming neither much larger than $(1 + \epsilon)\pi_{\theta_k}(a_i|s_i)$ nor much smaller than $(1 - \epsilon)\pi_{\theta_k}(a_i|s_i)$ for $i = 1, \ldots, m$. In this subsection, we consider the constraint function

$$\eta(\pi, \pi_{\theta_k}) = \max_{i=1,\ldots,m} \frac{|\pi(a_i|s_i) - \pi_{\theta_k}(a_i|s_i)|}{\pi_{\theta_k}(a_i|s_i)} \tag{23}$$

which leads us to the following optimization problem:

$$\underset{\pi(a_1|s_1),\ldots,\pi(a_m|s_m)}{\text{maximize}} \quad \sum_{i=1}^m A^{\pi_{\theta_k}}(s_i, a_i) \frac{\pi(a_i|s_i)}{\pi_k(a_i|s_i)} \tag{24}$$

$$\text{subject to} \quad \left| \frac{\pi(a_i|s_i) - \pi_{\theta_k}(a_i|s_i)}{\pi_{\theta_k}(a_i|s_i)} \right| \leq \epsilon \quad i = 1, \ldots, m \tag{25}$$

$$\sum_{i=1}^m \left( \frac{\pi(a_i|s_i) - \pi_{\theta_k}(a_i|s_i)}{\pi_{\theta_k}(a_i|s_i)} \right)^2 \leq \delta \tag{26}$$

Note that here we are using a variation of the SPU methodology described in Section 4 since here we first create estimates of the expectations in the objective and constraints and then solve the optimization problem (rather than first solve the optimization problem and then take samples as done for Theorems 1 and 2). Note that we have also included an aggregated constraint (26) in addition to the PPO-like constraint (25), which further ensures that the updated policy is close to $\pi_{\theta_k}$.

**Theorem 3** *The optimal solution to the optimization problem (24-26) is given by:*

$$\pi^*(a_i|s_i) = \begin{cases} \pi_{\theta_k}(a_i|s_i) \min\{1 + \lambda A_i, 1 + \epsilon\} & A_i \geq 0 \\ \pi_{\theta_k}(a_i|s_i) \max\{1 + \lambda A_i, 1 - \epsilon\} & A_i < 0 \end{cases} \tag{27}$$

*for some $\lambda > 0$ where $A_i \triangleq A^{\pi_{\theta_k}}(s_i, a_i)$ (Proof in subsection A.3).*

To simplify the algorithm, we treat $\lambda$ as a hyper-parameter rather than $\delta$. After solving for $\pi^*$, we seek a parameterized policy $\pi_\theta$ that is close to $\pi^*$ by minimizing their mean square error over sampled states and actions, i.e. by updating $\theta$ in the negative direction of $\nabla_\theta \sum_i (\pi_\theta(a_i|s_i) - \pi^*(a_i|s_i))^2$. This loss is used for supervised training instead of the KL because we take estimates before forming the optimization problem. Thus, the optimal values for the decision variables do not completely characterize a distribution. We refer to this approach as SPU with the $L^\infty$ constraint.

Although we consider three classes of proximity constraint, there may be yet another class that leads to even better performance. The methodology allows researchers to explore other proximity constraints in the future.

## 6 EXPERIMENTAL RESULTS

Extensive experimental results demonstrate SPU outperforms recent state-of-the-art methods for environments with continuous or discrete action spaces. We provide ablation studies to show the importance of the different algorithmic components, and a sensitivity analysis to show that SPU's performance is relatively insensitive to hyper-parameter choices. There are two definitions we use to conclude A is more sample efficient than B: (i) A takes fewer environment interactions to achieve a pre-defined performance threshold (Kakade, 2003); (ii) the averaged final performance of A is higher than that of B given the same number environment interactions (Schulman et al., 2017). Implementation details are provided in Appendix D.

### 6.1 RESULTS ON MUJOCO

The Mujoco (Todorov et al., 2012) simulated robotics environments provided by OpenAI gym (Brockman et al., 2016) have become a popular benchmark for control problems with continuous action spaces. In terms of final performance averaged over all available ten Mujoco environments and ten different seeds in each, SPU with $L^\infty$ constraint (Section 5.3) and SPU with forward KL constraints (Section 5.1) outperform TRPO by $6\%$ and $27\%$ respectively. Since the forward-KL approach is our best performing approach, we focus subsequent analysis on it and hereafter refer to it as SPU. SPU also outperforms PPO by $17\%$. Figure 1 illustrates the performance of SPU versus TRPO, PPO.

To ensure that SPU is not only better than TRPO in terms of performance gain early during training, we further retrain both policies for 3 million timesteps. Again here, SPU outperforms TRPO by $28\%$. Figure 3 in the Appendix illustrates the performance for each environment. Code for the Mujoco experiments is at `https://github.com/quanvuong/Supervised_Policy_Update`.

### 6.2 ABLATION STUDIES FOR MUJOCO

The indicator variable in (18) enforces the disaggregated constraint. We refer to it as *per-state acceptance*. Removing this component is equivalent to removing the indicator variable. We refer to using $\sum_i D_{\mathrm{KL}}(\pi_\theta \parallel \pi_{\theta_k})[s_i]$ to determine the number of training epochs as *dynamic stopping*. Without this component, the number of training epochs is a hyper-parameter. We also tried removing $\nabla_\theta D_{\mathrm{KL}}(\pi_\theta \parallel \pi_{\theta_k})[s_i]$ from the gradient update step in (18). Table 1 illustrates the contribution of the different components of SPU to the overall performance. The third row shows that the term $\nabla_\theta D_{\mathrm{KL}}(\pi_\theta \parallel \pi_{\theta_k})[s_i]$ makes a crucially important contribution to SPU. Furthermore, per-state acceptance and dynamic stopping are both also important for obtaining high performance, with the former playing a more central role. When a component is removed, the hyper-parameters are retuned to ensure that the best possible performance is obtained with the alternative (simpler) algorithm.

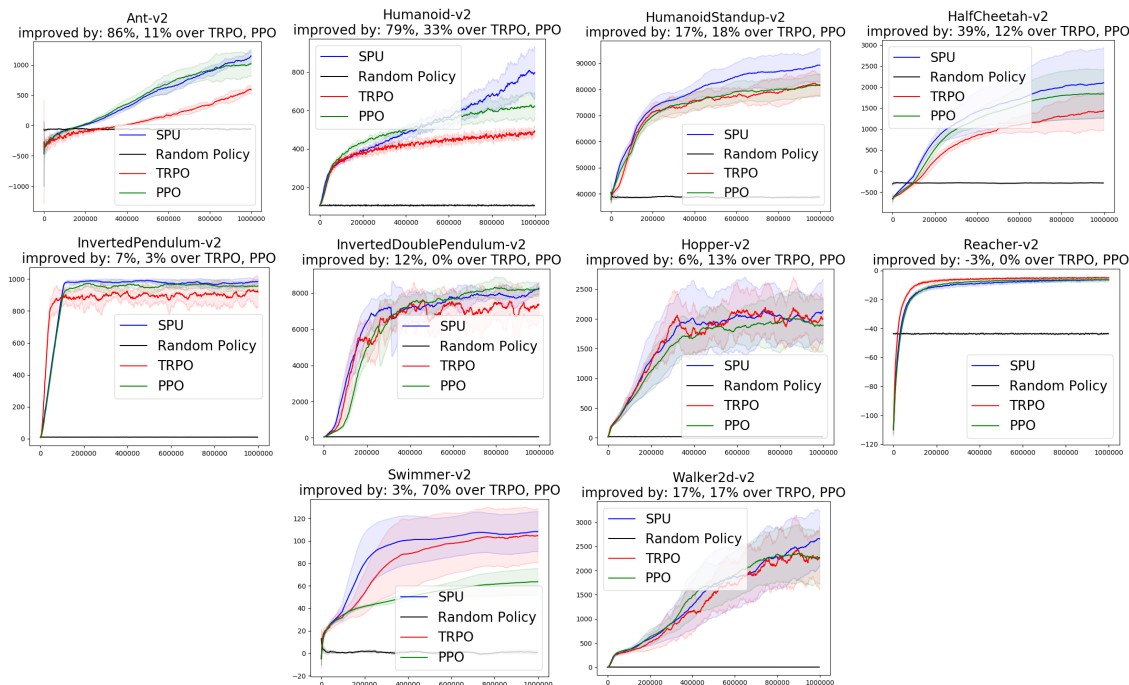

Figure 1: SPU versus TRPO, PPO on 10 Mujoco environments in 1 million timesteps. The x-axis indicates timesteps. The y-axis indicates the average episode reward of the last 100 episodes.

Table 1: Ablation study for SPU

| Approach | Percentage better than TRPO | Performance vs. original algorithm |
|---|---|---|
| Original Algorithm | 27% | 0% |
| No grad KL | 4% | - 85% |
| No dynamic stopping | 24% | - 11% |
| No per-state acceptance | 9% | - 67% |

## 6.3 SENSITIVITY ANALYSIS ON MUJOCO

To demonstrate the practicality of SPU, we show that its high performance is insensitive to hyper-parameter choice. One way to show this is as follows: for each SPU hyper-parameter, select a reasonably large interval, randomly sample the value of the hyper parameter from this interval, and then compare SPU (using the randomly chosen hyper-parameter values) with TRPO. We sampled 100 SPU hyper-parameter vectors (each vector including $\delta, \epsilon, \lambda$), and for each one determined the relative performance with respect to TRPO. First, we found that for all 100 random hyper-parameter value samples, SPU performed better than TRPO. 75% and 50% of the samples outperformed TRPO by at least 18% and 21% respectively. The full CDF is given in Figure 4 in the Appendix. We can conclude that SPU's superior performance is largely insensitive to hyper-parameter values.

## 6.4 RESULTS ON ATARI

(Rajeswaran et al., 2017; Mania et al., 2018) demonstrates that neural networks are not needed to obtain high performance in many Mujoco environments. To conclusively evaluate SPU, we compare it against PPO on the Arcade Learning Environments (Bellemare et al., 2012) exposed through OpenAI gym (Brockman et al., 2016). Using the same network architecture and hyper-parameters, we learn to play 60 Atari games from raw pixels and rewards. This is highly challenging because of the diversity in the games and the high dimensionality of the observations.

Here, we compare SPU against PPO because PPO outperforms TRPO by $9\%$ in Mujoco. Averaged over 60 Atari environments and 20 seeds, SPU is $\mathbf{55\%}$ better than PPO in terms of averaged final performance. Figure 2 provides a high-level overview of the result. The dots in the shaded area represent environments where their performances are roughly similar. The dots to the right of the shaded area represent environment where SPU is more sample efficient than PPO. We can draw two conclusions: (i) In 36 environments, SPU and PPO perform roughly the same ; SPU clearly outperforms PPO in 15 environments while PPO clearly outperforms SPU in 9; (ii) In those 15+9 environments, the extent to which SPU outperforms PPO is much larger than the extent to which PPO outperforms SPU. Figure 5, Figure 6 and Figure 7 in the Appendix illustrate the performance of SPU vs PPO throughout training. SPU's high performance in both the Mujoco and Atari domains demonstrates its high performance and generality.

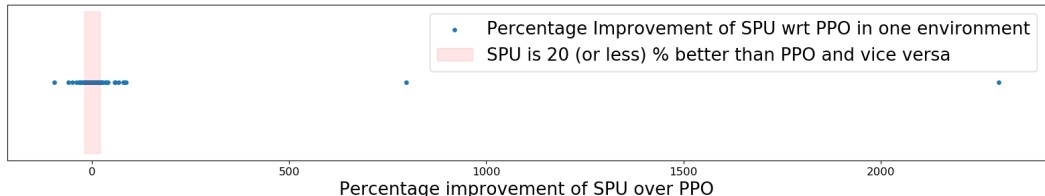

Figure 2: High-level overview of results on Atari

# 7 ACKNOWLEDGEMENTS

We would like to acknowledge the extremely helpful support by the NYU Shanghai High Performance Computing Administrator Zhiguo Qi. We also are grateful to OpenAI for open-sourcing their baselines codes.

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

# Appendices

## A    Proofs for non-parameterized optimization problems

### A.1    Forward KL Aggregated and Disaggregated Constraints

We first show that (12)-(14) is a convex optimization. To this end, first note that the objective (12) is a linear function of the decision variables $\pi = \{\pi(a|s) : s \in \mathcal{S}, a \in \mathcal{A}\}$. The LHS of (14) can be rewritten as: $\sum_{a \in \mathcal{A}} \pi(a|s) \log \pi(a|s) - \sum_{a \in \mathcal{A}} \pi(a|s) \log \pi_{\theta_k}(a|s)$. The second term is a linear function of $\pi$. The first term is a convex function since the second derivative of each summand is always positive. The LHS of (14) is thus a convex function. By extension, the LHS of (13) is also a convex function since it is a nonnegative weighted sum of convex functions. The problem (12)-(14) is thus a convex optimization problem. According to Slater's constraint qualification, strong duality holds since $\pi_{\theta_k}$ is a feasible solution to (12)-(14) where the inequality holds strictly.

We can therefore solve (12)-(14) by solving the related Lagrangian problem. For a fixed $\lambda$ consider:

$$\underset{\pi \in \Pi}{\text{maximize}} \quad \sum_s d^{\pi_{\theta_k}}(s)\{\underset{a \sim \pi(\cdot|s)}{\mathbb{E}}[A^{\pi_{\theta_k}}(s,a)] - \lambda D_{\text{KL}}(\pi \parallel \pi_{\theta_k})[s]\} \tag{28}$$

$$\text{subject to} \quad D_{\text{KL}}(\pi \parallel \pi_{\theta_k})[s] \leq \epsilon \text{ for all } s \tag{29}$$

The above problem decomposes into separate problems, one for each state $s$:

$$\underset{\pi(\cdot|s)}{\text{maximize}} \quad \underset{a \sim \pi(\cdot|s)}{\mathbb{E}}[A^{\pi_{\theta_k}}(s,a)] - \lambda D_{\text{KL}}(\pi \parallel \pi_{\theta_k})[s] \tag{30}$$

$$\text{subject to} \quad D_{\text{KL}}(\pi \parallel \pi_{\theta_k})[s] \leq \epsilon \tag{31}$$

Further consider the unconstrained problem (30) without the constraint (31):

$$\underset{\pi(\cdot|s)}{\text{maximize}} \quad \sum_{a=1}^{K} \pi(a|s) \left[ A^{\pi_{\theta_k}}(s,a) - \lambda \log \left( \frac{\pi(a|s)}{\pi_{\theta_k}(a|s)} \right) \right] \tag{32}$$

$$\text{subject to} \quad \sum_{a=1}^{K} \pi(a|s) = 1 \tag{33}$$

$$\pi(a|s) \geq 0, \quad a = 1, \ldots, K \tag{34}$$

A simple Lagrange-multiplier argument shows that the opimal solution to (32)-(34) is given by:

$$\pi^{\lambda}(a|s) = \frac{\pi_{\theta_k}(a|s)}{Z_{\lambda}(s)} e^{A^{\pi_{\theta_k}}(s,a)/\lambda}$$

where $Z_{\lambda}(s)$ is defined so that $\pi^{\lambda}(\cdot|s)$ is a valid distribution. Now returning to the decomposed constrained problem (30)-(31), there are two cases to consider. The first case is when $D_{\text{KL}}(\pi^{\lambda} \parallel \pi_{\theta_k})[s] \leq \epsilon$. In this case, the optimal solution to (30)-(31) is $\pi^{\lambda}(a|s)$. The second case is when $D_{\text{KL}}(\pi^{\lambda} \parallel \pi_{\theta_k})[s] > \epsilon$. In this case the optimal is $\pi^{\lambda}(a|s)$ with $\lambda$ replaced with $\lambda_s$, where $\lambda_s$ is the solution to $D_{\text{KL}}(\pi^{\lambda} \parallel \pi_{\theta_k})[s] = \epsilon$. Thus, an optimal solution to (30)-(31) is given by:

$$\tilde{\pi}^{\lambda}(a|s) = \begin{cases} \dfrac{\pi_{\theta_k}(a|s)}{Z(s)} e^{A^{\pi_{\theta_k}}(s,a)/\lambda} & s \in \Gamma^{\lambda} \\ \dfrac{\pi_{\theta_k}(a|s)}{Z(s)} e^{A^{\pi_{\theta_k}}(s,a)/\lambda_s} & s \notin \Gamma^{\lambda} \end{cases} \tag{35}$$

where $\Gamma^{\lambda} = \{s : D_{\text{KL}}(\pi^{\lambda} \parallel \pi_{\theta_k})[s] \leq \epsilon\}$.

To find the Lagrange multiplier $\lambda$, we can then do a line search to find the $\lambda$ that satisfies:

$$\sum_s d^{\pi_{\theta_k}}(s) D_{\text{KL}}(\tilde{\pi}^{\lambda} \parallel \pi_{\theta_k})[s] = \delta \tag{36}$$

$\square$

## A.2 BACKWARD KL CONSTRAINT

The problem (19)-(20) decomposes into separate problems, one for each state $s \in \mathcal{S}$:

$$\underset{\pi(\cdot|s)}{\text{maximize}} \quad \underset{a \sim \pi_{\theta_k}(\cdot|s)}{\mathbb{E}} \left[ \frac{\pi(a|s)}{\pi_{\theta_k}(a|s)} A^{\pi_{\theta_k}}(s,a) \right] \tag{37}$$

$$\text{subject to} \quad \underset{a \sim \pi_{\theta_k}(\cdot|s)}{\mathbb{E}} \left[ \log \frac{\pi_{\theta_k}(a|s)}{\pi(a|s)} \right] \leq \epsilon \tag{38}$$

After some algebra, we see that above optimization problem is equivalent to:

$$\underset{\pi(\cdot|s)}{\text{maximize}} \quad \sum_{a=1}^{K} A^{\pi_{\theta_k}}(s,a)\pi(a|s) \tag{39}$$

$$\text{subject to} \quad -\sum_{a=1}^{K} \pi_{\theta_k}(a|s) \log \pi(a|s) \leq \epsilon' \tag{40}$$

$$\sum_{a=1}^{K} \pi(a|s) = 1 \tag{41}$$

$$\pi(a|s) \geq 0, \quad a = 1, \dots, K \tag{42}$$

where $\epsilon' = \epsilon + entropy(\pi_{\theta_k})$. (39)-(42) is a convex optimization problem with Slater's condition holding. Strong duality thus holds for the problem (39)-(42). Applying standard Lagrange multiplier arguments, it is easily seen that the solution to (39)-(42) is

$$\pi^*(a|s) = \pi_{\theta_k}(a|s)\frac{\lambda(s)}{\lambda'(s) - A^{\pi_{\theta_k}}(s,a)}$$

where $\lambda(s)$ and $\lambda'(s)$ are constants chosen such that the disaggregegated KL constraint is binding and the sum of the probabilities equals 1. It is easily seen $\lambda(s) > 0$ and $\lambda'(s) > \max_a A^{\pi_{\theta_k}}(s,a)$ $\square$

## A.3 $L^\infty$ CONSTRAINT

The problem (24-26) is equivalent to:

$$\underset{\pi(a_1|s_1),\dots,\pi(a_m|s_m)}{\text{maximize}} \quad \sum_{i=1}^{m} A^{\pi_{\theta_k}}(s_i,a_i)\frac{\pi(a_i|s_i)}{\pi_k(a_i|s_i)} \tag{43}$$

$$\text{subject to} \quad 1 - \epsilon \leq \frac{\pi(a_i|s_i)}{\pi_{\theta_k}(a_i|s_i)} \leq 1 + \epsilon \quad i = 1, \dots, m \tag{44}$$

$$\sum_{i=1}^{m} \left( \frac{\pi(a_i|s_i) - \pi_{\theta_k}(a_i|s_i)}{\pi_{\theta_k}(a_i|s_i)} \right)^2 \leq \delta \tag{45}$$

This problem is clearly convex. $\pi_{\theta_k}(a_i|s_i), i = 1, \dots, m$ is a feasible solution where the inequality constraint holds strictly. Strong duality thus holds according to Slater's constraint qualification. To solve (43)-(45), we can therefore solve the related Lagrangian problem for fixed $\lambda$:

$$\underset{\pi(a_1|s_1),\dots,\pi(a_m|s_m)}{\text{maximize}} \quad \sum_{i=1}^{m} \left[ A^{\pi_{\theta_k}}(s_i,a_i)\frac{\pi(a_i|s_i)}{\pi_k(a_i|s_i)} - \lambda\left( \frac{\pi(a_i|s_i) - \pi_{\theta_k}(a_i|s_i)}{\pi_{\theta_k}(a_i|s_i)} \right)^2 \right] \tag{46}$$

$$\text{subject to} \quad 1 - \epsilon \leq \frac{\pi(a_i|s_i)}{\pi_{\theta_k}(a_i|s_i)} \leq 1 + \epsilon \quad i = 1, \dots, m \tag{47}$$

which is separable and decomposes into m separate problems, one for each $s_i$:

$$\underset{\pi(a_i|s_i)}{\text{maximize}} \quad A^{\pi_{\theta_k}}(s_i,a_i)\frac{\pi(a_i|s_i)}{\pi_k(a_i|s_i)} - \lambda\left( \frac{\pi(a_i|s_i) - \pi_{\theta_k}(a_i|s_i)}{\pi_{\theta_k}(a_i|s_i)} \right)^2 \tag{48}$$

$$\text{subject to} \quad 1 - \epsilon \leq \frac{\pi(a_i|s_i)}{\pi_{\theta_k}(a_i|s_i)} \leq 1 + \epsilon \tag{49}$$

The solution to the unconstrained problem (48) without the constraint (49) is:

$$\pi^*(a_i|s_i) = \pi_{\theta_k}(a_i|s_i)\left(1 + \frac{A^{\pi_{\theta_k}}(s_i, a_i)}{2\lambda}\right)$$

Now consider the constrained problem (48)-(49). If $A^{\pi_{\theta_k}}(s_i, a_i) \geq 0$ and $\pi^*(a_i|s_i) > \pi_{\theta_k}(a_i|s_i)(1+\epsilon)$, the optimal solution is $\pi_{\theta_k}(a_i|s_i)(1+\epsilon)$. Similarly, If $A^{\pi_{\theta_k}}(s_i, a_i) < 0$ and $\pi^*(a_i|s_i) < \pi_{\theta_k}(a_i|s_i)(1-\epsilon)$, the optimal solution is $\pi_{\theta_k}(a_i|s_i)(1-\epsilon)$. Rearranging the terms gives Theorem 3. To obtain $\lambda$, we can perform a line search over $\lambda$ so that the constraint (45) is binding. $\square$

# B   DERIVATIONS THE GRADIENT OF LOSS FUNCTION FOR SPU

Let $CE$ stands for CrossEntropy.

## B.1   FORWARD-KL

$CE(\pi_\theta || \tilde{\pi}^{\tilde{\lambda}_s})[s]$

$= -\sum_a \pi_\theta(a|s) \log \tilde{\pi}^{\tilde{\lambda}_s}(a|s)$(Expanding the definition of cross entropy)

$= -\sum_a \pi_\theta(a|s) \log \left( \dfrac{\pi_{\theta_k}(a|s)}{Z_{\tilde{\lambda}_s}(s)} e^{A^{\pi_{\theta_k}}(s,a)/\tilde{\lambda}_s} \right)$ (Expanding the definition of $\tilde{\pi}^{\tilde{\lambda}_s}$)

$= -\sum_a \pi_\theta(a|s) \log \left( \dfrac{\pi_{\theta_k}(a|s)}{Z_{\tilde{\lambda}_s}(s)} \right) - \sum_a \pi_\theta(a|s) \dfrac{A^{\pi_{\theta_k}}(s,a)}{\tilde{\lambda}_s}$(Log of product is sum of log)

$= -\sum_a \pi_\theta(a|s) \log \pi_{\theta_k}(a|s) + \sum_a \pi_\theta(a|s) \log Z_{\tilde{\lambda}_s}(s) - \dfrac{1}{\tilde{\lambda}_s} \sum_a \pi_{\theta_k}(a|s) \dfrac{\pi_\theta(a|s)}{\pi_{\theta_k}(a|s)} A^{\pi_{\theta_k}}(s,a)$

$= CE(\pi_\theta || \pi_{\theta_k})[s] + \log Z_{\tilde{\lambda}_s}(s) - \dfrac{1}{\tilde{\lambda}_s} \underset{a \sim \pi_{\theta_k}(.|s)}{E} \left[ \dfrac{\pi_\theta(a|s)}{\pi_{\theta_k}(a|s)} A^{\pi_{\theta_k}}(s,a) \right]$

$\Rightarrow \nabla_\theta CE(\pi_\theta || \tilde{\pi}^{\tilde{\lambda}_s})[s] = \nabla_\theta CE(\pi_\theta || \pi_{\theta_k})[s] - \dfrac{1}{\tilde{\lambda}_s} \underset{a \sim \pi_{\theta_k}(.|s)}{E} \left[ \dfrac{\nabla_\theta \pi_\theta(a|s)}{\pi_{\theta_k}(a|s)} A^{\pi_{\theta_k}}(s,a) \right]$ (Taking gradient on both sides)

$\Rightarrow \nabla_\theta D_{\text{KL}}(\pi_\theta \| \tilde{\pi}^{\tilde{\lambda}_s})[s] = \nabla_\theta D_{\text{KL}}(\pi_\theta \| \pi_{\theta_k})[s] - \dfrac{1}{\tilde{\lambda}_s} \underset{a \sim \pi_{\theta_k}(.|s)}{E} \left[ \dfrac{\nabla_\theta \pi_\theta(a|s)}{\pi_{\theta_k}(a|s)} A^{\pi_{\theta_k}}(s,a) \right]$

(Adding the gradient of the entropy on both sides and collapse the sum of gradients of cross entropy and entropy into the gradient of the KL)

## B.2   REVERSE-KL

$CE(\pi_\theta || \pi^*)[s]$

$= -\sum_a \pi_\theta(a|s) \log \pi^*(a|s)$(Expanding the definition of cross entropy)

$= -\sum_a \pi_\theta(a|s) \log \left( \pi_{\theta_k}(a|s) \dfrac{\lambda(s)}{\lambda'(s) - A^{\pi_{\theta_k}}(s,a)} \right)$ (Expanding the definition of $\pi^*$)

$= -\sum_a \pi_\theta(a|s) \log \pi_{\theta_k}(a|s) - \sum_a \pi_\theta(a|s) \log \lambda(s) + \sum_a \pi_\theta(a|s) \log(\lambda'(s) - A^{\pi_{\theta_k}}(s,a))$

$= CE(\pi_\theta || \pi_{\theta_k})[s] - \lambda(s) + E_{a \sim \pi_{\theta_k}} \left[ \dfrac{\pi_\theta(a|s)}{\pi_{\theta_k}(a|s)} \log(\lambda'(s) - A^{\pi_{\theta_k}}(s,a)) \right]$

$= CE(\pi_\theta || \pi_{\theta_k})[s] - \lambda(s) - E_{a \sim \pi_{\theta_k}} \left[ \dfrac{\pi_\theta(a|s)}{\pi_{\theta_k}(a|s)} \log \dfrac{1}{\lambda'(s) - A^{\pi_{\theta_k}}(s,a)} \right]$

$\Rightarrow \nabla_\theta CE(\pi_\theta || \pi^*)[s] = \nabla_\theta CE(\pi_\theta || \pi_{\theta_k})[s] - E_{a \sim \pi_{\theta_k}} \left[ \dfrac{\nabla_\theta \pi_\theta(a|s)}{\pi_{\theta_k}(a|s)} \log \dfrac{1}{\lambda'(s) - A^{\pi_{\theta_k}}(s,a)} \right]$ (Taking gradient on both sides)

$\Rightarrow \nabla_\theta D_{\text{KL}}(\pi_\theta \| \pi^*)[s] = \nabla_\theta D_{\text{KL}}(\pi_\theta \| \pi_{\theta_k})[s] - E_{a \sim \pi_{\theta_k}} \left[ \dfrac{\nabla_\theta \pi_\theta(a|s)}{\pi_{\theta_k}(a|s)} \log \dfrac{1}{\lambda'(s) - A^{\pi_{\theta_k}}(s,a)} \right]$

(Adding the gradient of the entropy on both sides and collapse the sum of gradients of cross entropy and entropy into the gradient of the KL)

## C  EXTENSION TO CONTINUOUS STATE AND ACTION SPACES

The methodology developed in the body of this paper also applies to continuous state and action spaces. In this section, we outline the modifications that are necessary for the continuous case.

We first modify the definition of $d^\pi(s)$ by replacing $P_\pi(s_t = s)$ with $\frac{d}{ds}P_\pi(s_t \leq s)$ so that $d^\pi(s)$ becomes a density function over the state space. With this modification, the definition of $\bar{D}_{\mathrm{KL}}(\pi \parallel \pi_k)$ and the approximation (8) are unchanged. The SPU framework described in Section 4 is also unchanged.

Consider now the non-parameterized optimization problem with aggregate and disaggregate constraints (12-14), but with continuous state and action space:

$$\underset{\pi \in \Pi}{\text{maximize}} \quad \int d^{\pi_{\theta_k}}(s) \underset{a \sim \pi(\cdot|s)}{\mathbb{E}}[A^{\pi_{\theta_k}}(s,a)]ds \tag{50}$$

$$\text{subject to} \quad \int d^{\pi_{\theta_k}}(s) D_{\mathrm{KL}}(\pi \parallel \pi_{\theta_k})[s]ds \leq \delta \tag{51}$$

$$D_{\mathrm{KL}}(\pi \parallel \pi_{\theta_k})[s] \leq \epsilon \text{ for all } s \tag{52}$$

Theorem 1 holds although its proof needs to be slightly modified as follows. It is straightforward to show that (50-52) remains a convex optimization problem. We can therefore solve (50-52) by solving the Lagrangian (28-29) with the sum replaced with an integral. This problem again decomposes with separate problems for each $s \in \mathcal{S}$ giving exactly the same equations (30-31). The proof then proceeds as in the remainder of the proof of Theorem 1.

Theorem 2 and 3 are also unchanged for continuous action spaces. Their proofs require slight modifications, as in the proof of Theorem 1.

## D  IMPLEMENTATION DETAILS AND HYPERPARAMETERS

### D.1  MUJOCO

As in (Schulman et al., 2017), for Mujoco environments, the policy is parameterized by a fully-connected feed-forward neural network with two hidden layers, each with 64 units and tanh nonlinearities. The policy outputs the mean of a Gaussian distribution with state-independent variable standard deviations, following (Schulman et al., 2015a; Duan et al., 2016). The action dimensions are assumed to be independent. The probability of an action is given by the multivariate Gaussian probability distribution function. The baseline used in the advantage value calculation is parameterized by a similarly sized neural network, trained to minimize the MSE between the sampled states TD$-\lambda$ returns and the their predicted values. For both the policy and baseline network, SPU and TRPO use the same architecture. To calculate the advantage values, we use Generalized Advantage Estimation (Schulman et al., 2015b). States are normalized by dividing the running mean and dividing by the running standard deviation before being fed to any neural networks. The advantage values are normalized by dividing the batch mean and dividing by the batch standard deviation before being used for policy update. The TRPO result is obtained by running the TRPO implementation provided by OpenAI (Dhariwal et al., 2017), commit 3cc7df060800a45890908045b79821a13c4babdb. At every iteration, SPU collects 2048 samples before updating the policy and the baseline network. For both networks, gradient descent is performed using Adam (Kingma & Ba, 2014) with step size 0.0003, minibatch size of 64. The step size is linearly annealed to 0 over the course of training. $\gamma$ and $\lambda$ for GAE (Schulman et al., 2015b) are set to 0.99 and 0.95 respectively. For SPU, $\delta, \epsilon, \lambda$ and the maximum number of epochs per iteration are set to 0.05/1.2, 0.05, 1.3 and 30 respectively. Training is performed for 1 million timesteps for both SPU and PPO. In the sensitivity analysis, the ranges of values for the hyper-parameters $\delta, \epsilon, \lambda$ and maximum number of epochs are $[0.05, 0.07]$, $[0.01, 0.07]$, $[1.0, 1.2]$ and $[5, 30]$ respectively.

## D.2 ATARI

Unless otherwise mentioned, the hyper-parameter values are the same as in subsection D.1. The policy is parameterized by a convolutional neural network with the same architecture as described in Mnih et al. (2015). The output of the network is passed through a relu, linear and softmax layer in that order to give the action distribution. The output of the network is also passed through a different linear layer to give the baseline value. States are normalized by dividing by 255 before being fed into any network. The TRPO result is obtained by running the PPO implementation provided by OpenAI (Dhariwal et al., 2017), commit 3cc7df060800a45890908045b79821a13c4babdb. 8 different processes run in parallel to collect timesteps. At every iteration, each process collects 256 samples before updating the policy and the baseline network. Each process calculates its own update to the network's parameters and the updates are averaged over all processes before being used to update the network's parameters. Gradient descent is performed using Adam (Kingma & Ba, 2014) with step size 0.0001. In each process, random number generators are initialized with a different seed according to the formula $process\_seed = experiment\_seed + 10000 * process\_rank$. Training is performed for 10 million timesteps for both SPU and PPO. For SPU, $\delta, \epsilon, \lambda$ and the maximum number of epochs per iteration are set to 0.02, $\delta/1.3$, 1.1 and 9 respectively.

# E ALGORITHMIC DESCRIPTION FOR SPU

---

**Algorithm 1** Algorithmic description of forward-KL non-parameterized SPU

---

**Require:** A neural net $\pi_\theta$ that parameterizes the policy.
**Require:** A neural net $V_\phi$ that approximates $V^{\pi_\theta}$.
**Require:** General hyperparameters: $\gamma, \beta$ (advantage estimation using GAE), $\alpha$ (learning rate), N (number of trajectory per iteration), T (size of each trajectory), M (size of training minibatch).
**Require:** Algorithm-specific hyperparameters: $\delta$ (aggregated KL constraint), $\epsilon$ (disaggregated constraint), $\lambda, \zeta$ (max number of epoch).
1: **for** k = 1, 2, ... **do**
2:      under policy $\pi_{\theta_k}$, sample N trajectories, each of size T $(s_{it}, a_{it}, r_{it}, s_{i(t+1)})$, $i = 1, \ldots, N, t = 1, \ldots, T$
3:      Using any advantage value estimation scheme, estimate $A_{it}, \quad i = 1, \ldots, N, t = 1, \ldots, T$
4:      $\theta \leftarrow \theta_k$
5:      $\phi \leftarrow \phi_k$
6:      **for** $\zeta$ epochs **do**
7:          Sample M samples from the N trajectories, giving $\{s_1, a_1, A_1, \ldots, s_M, a_M, A_M\}$
8:          $L(\phi) = \frac{1}{M}\sum_m (V^{targ}(s_m) - V_\phi(s_m))^2$
9:          $\phi \leftarrow \phi - \alpha\nabla_\phi L(\phi)$
10:          $L(\theta) = \frac{1}{M}\sum_m \left[\nabla_\theta D_{KL}(\pi_\theta \parallel \pi_{\theta_k})[s_m] - \frac{1}{\lambda}\frac{\nabla_\theta \pi_\theta(a_m|s_m)}{\pi_{\theta_k}(a_m|s_m)}A_m\right]\mathbb{1}_{D_{KL}(\pi_\theta\parallel\pi_{\theta_k})[s_m]\leq\epsilon}$
11:          $\theta \leftarrow \theta - \alpha L(\theta)$
12:          **if** $\frac{1}{m}\sum_m D_{KL}(\pi \parallel \pi_{\theta_k})[s_m] > \delta$ **then**
13:              Break out of for loop
14:      $\theta_{k+1} \leftarrow \theta$
15:      $\phi_{k+1} \leftarrow \phi$

---

# F EXPERIMENTAL RESULTS

## F.1 RESULTS ON MUJOCO FOR 3 MILLION TIMESTEPS

TRPO and SPU were trained for 1 million timesteps to obtain the results in section 6. To ensure that SPU is not only better than TRPO in terms of performance gain early during training, we further retrain both policies for 3 million timesteps. Again here, SPU outperforms TRPO by 28%. Figure 3 illustrates the performance on each environment.

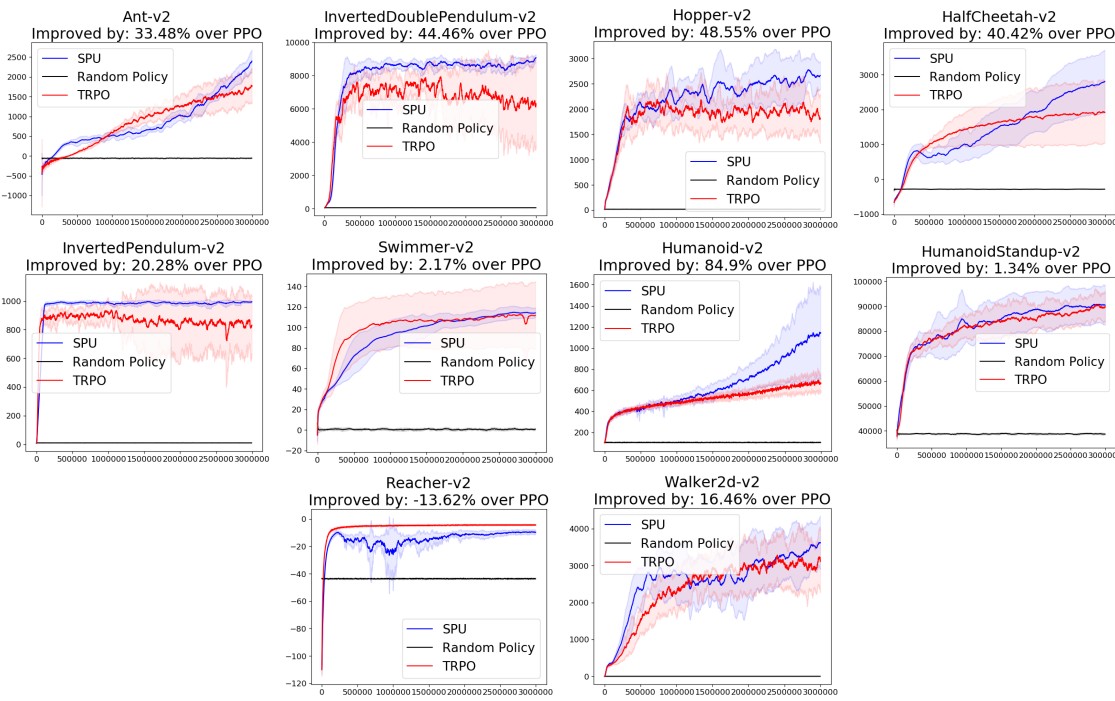

Figure 3: Performance of SPU versus TRPO on 10 Mujoco environments in 3 million timesteps. The x-axis indicates timesteps. The y-axis indicates the average episode reward of the last 100 episodes.

## F.2  SENSITIVITY ANALYSIS CDF FOR MUJOCO

When values for SPU hyper-parameter are randomly sampled as is explained in subsection 6.3, the percentage improvement of SPU over TRPO becomes a random variable. Figure 4 illustrates the CDF of this random variable.

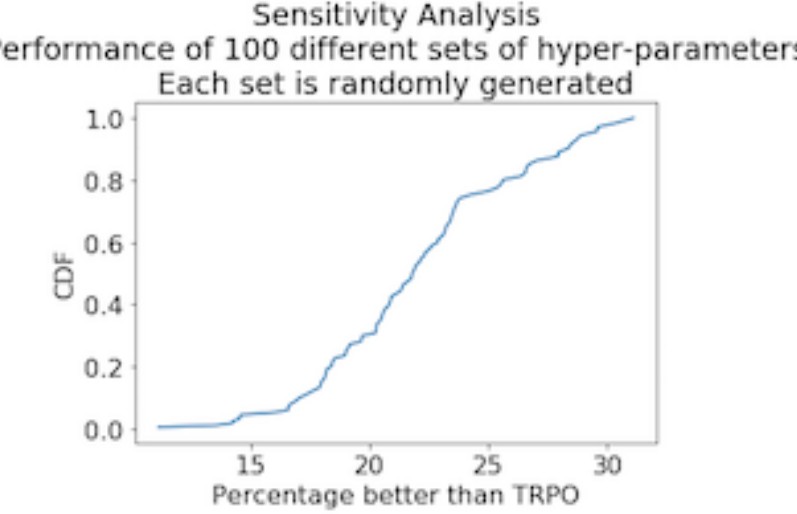

Figure 4: Sensitivity Analysis for SPU

## F.3    ATARI RESULTS

Figure 5, Figure 6 and Figure 7 illustrate the performance of SPU vs PPO throughout training in 60 Atari games.

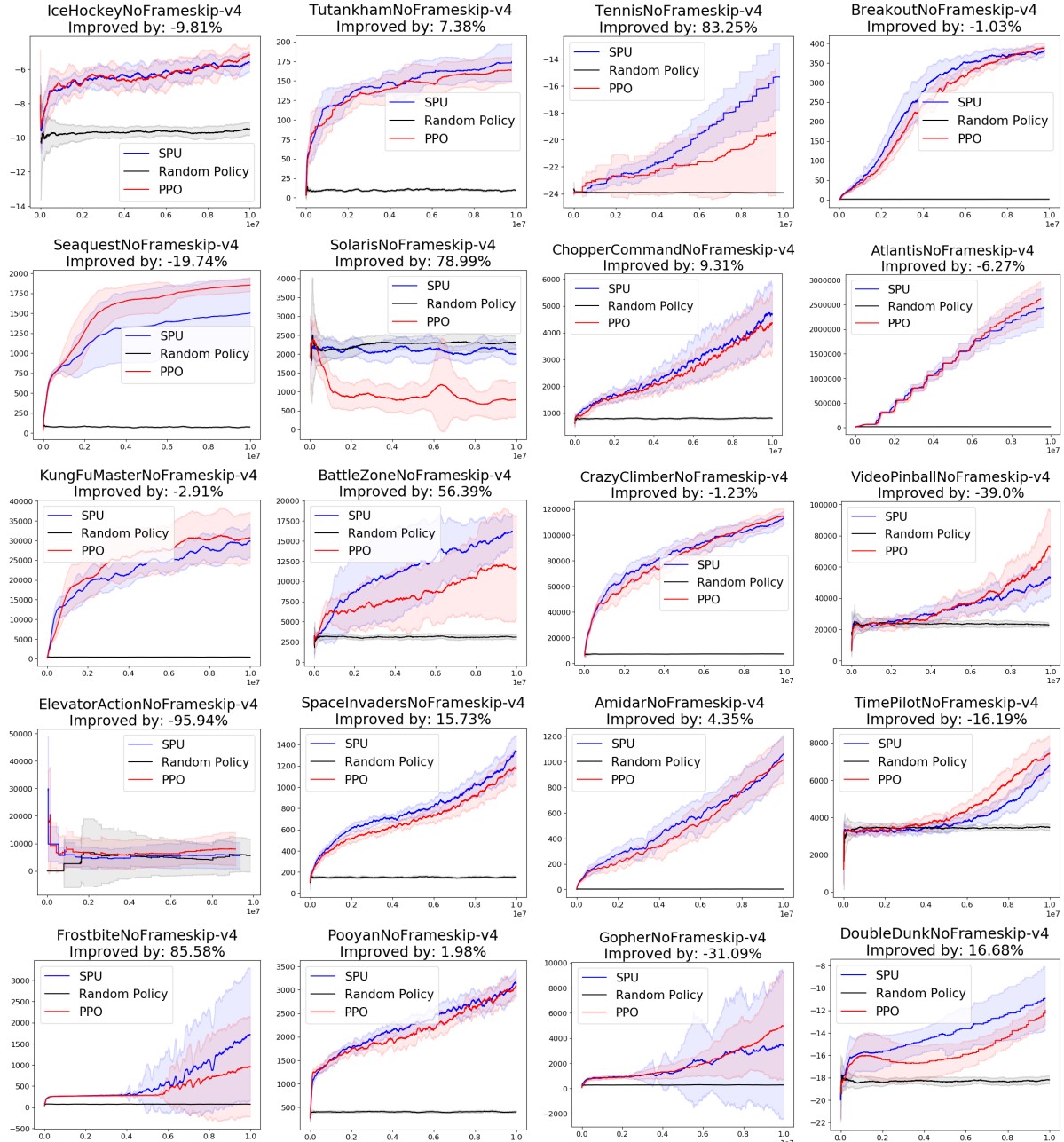

Figure 5: Atari results (Part 1) for SPU vs PPO. The x-axis indicates timesteps. The y-axis indicates the average episode reward of the last 100 episodes.

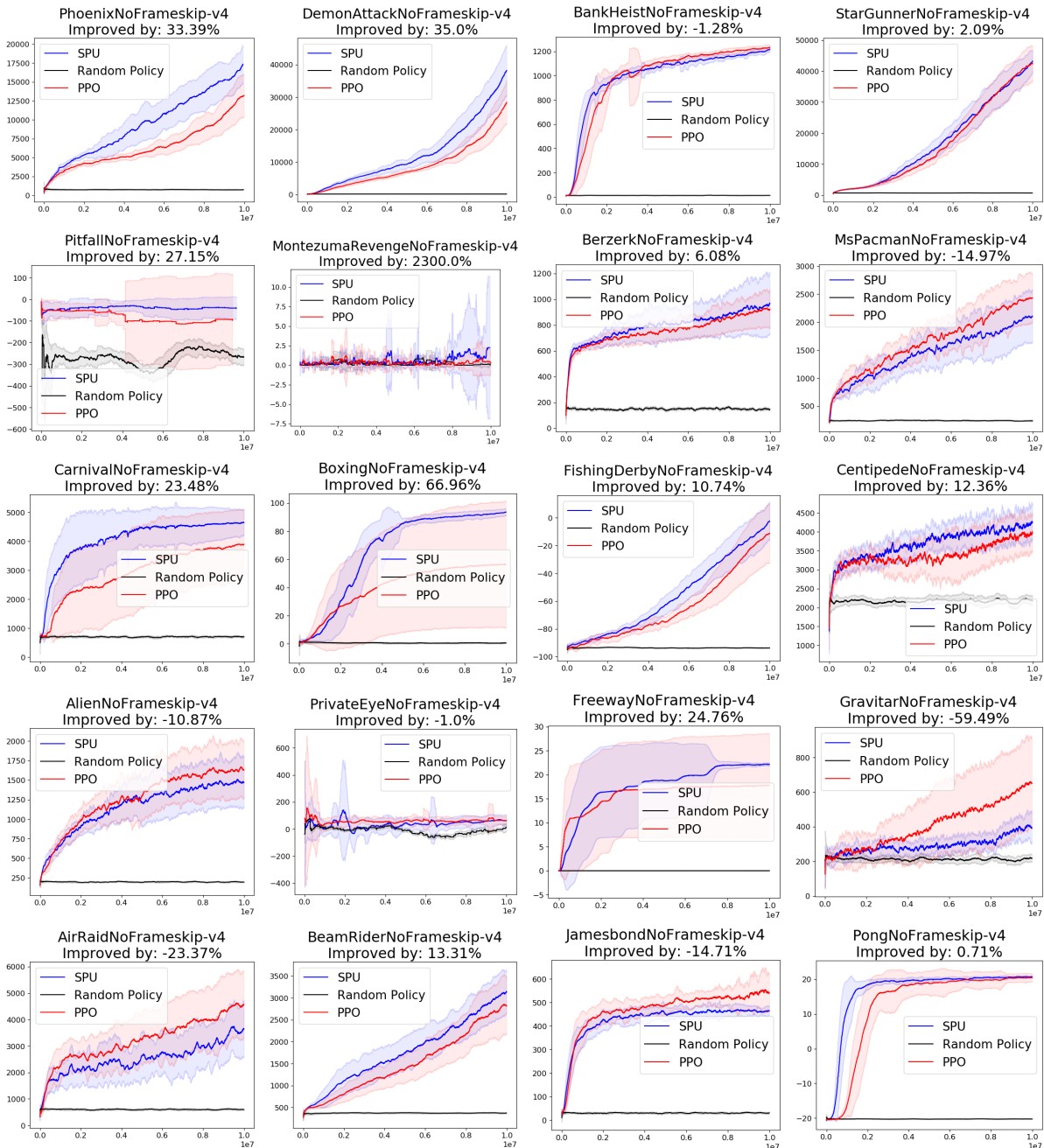

Figure 6: Atari results (Part 2) for SPU vs PPO. The x-axis indicates timesteps. The y-axis indicates the average episode reward of the last 100 episodes.

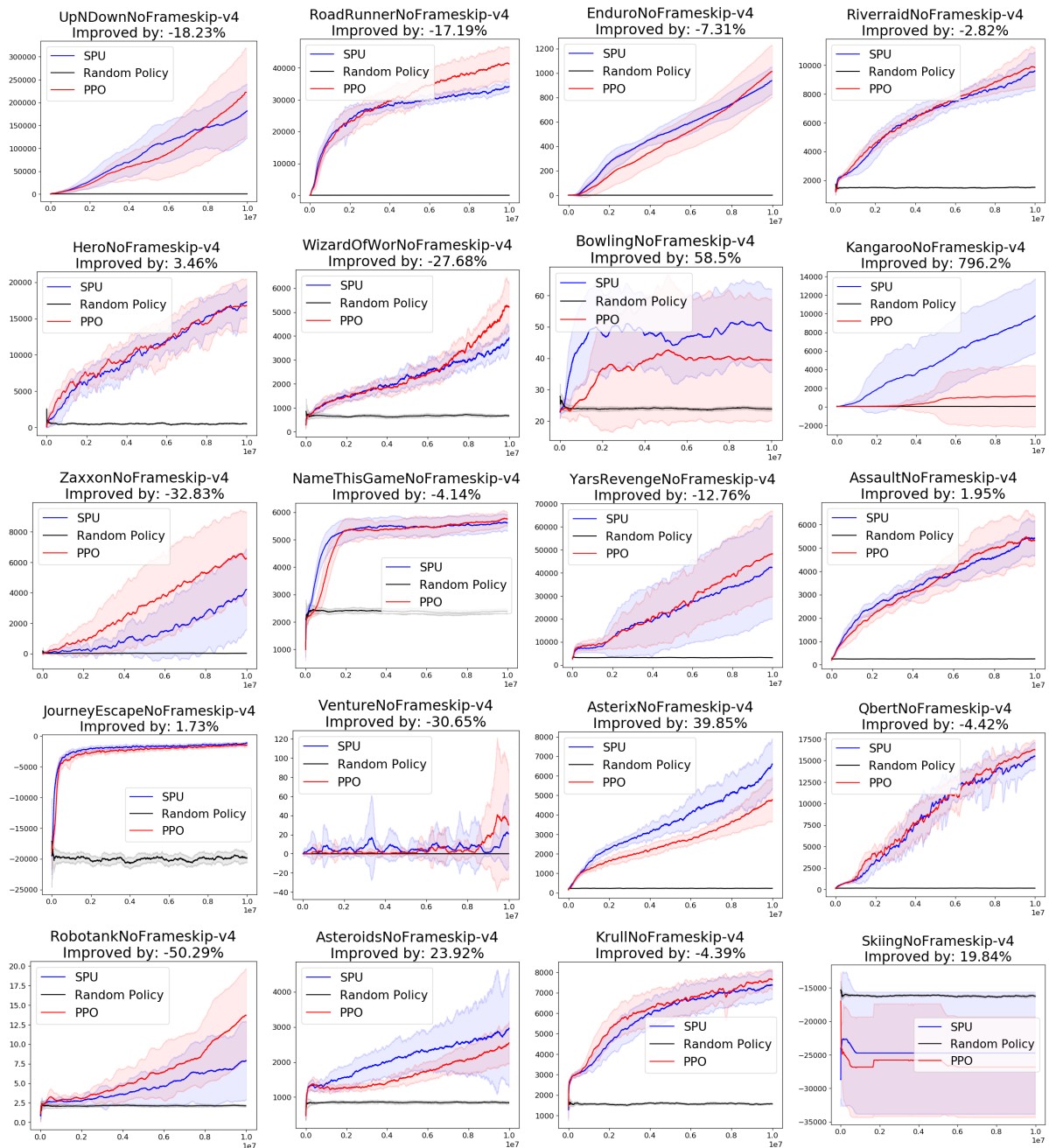

Figure 7: Atari results (Part 3) for SPU vs PPO. The x-axis indicates timesteps. The y-axis indicates the average episode reward of the last 100 episodes.

