# OpenReview forum: "Supervised Policy Update for Deep Reinforcement Learning"
_ICLR.cc/2019/Conference_

### Official Review · AnonReviewer1 · 2018-11-01
**SUPERVISED POLICY UPDATE**

**Rating:** 6
**Confidence:** 3

**Review:**

Overall this paper is ok. The algorithm seems novel, but is clearly very closely related to other things in the literature. The paper is also let down by poor exposition in several areas. The numerical results seem reasonably strong, at least against relatively old baselines.

Equation 8 is crucial to the final algorithm, but is presented with no proof or explanation.

Just above theorem 1 the sentence does not parse "Further, for each s, let λs be the solution to ", firstly there is no 'solution' to an equation, secondly should it be λs or pi?

The discussion following theorem 1 is very messy and hard to follow and the notation is horrendous. I'm confused as to why the indicator function in the 'disaggregated' update only includes states for which the constraint is already satisfied, what about the states where it is not? I presume this is because you initialize from the previous policy, but this seems very approximate and even worse updating the parameters for one state might significantly move the policy in some other state meaning large violations are possible and not dealt with.

The connections to the papers 'MAXIMUM A POSTERIORI POLICY OPTIMISATION' and 'Relative Entropy Policy Search' should be mentioned, as another commenter said previously.

I don't think TRPO/PPO is SOTA anymore, so maybe these baselines aren't particularly interesting.

Figure 2 is incomprehensible.

Two of the references are repeated (Schulman et al, Wang et al).

The appendices include long lists of equalities with no explanation (e.g. appendix B), how is a reader meant to reasonably follow those steps? Each non-trivial equality needs a sentence explaining what was used to get it.

---

> ### Author Response · Authors · 2018-11-10
> **Reply to your comment**
>
> Thank you so much for your detailed and thoughtful comments, we will revise the paper and figures based on your comments to better explain ourselves.
>
> Below, we address some of the specific concerns you have for our paper:
>
> “Equation 8 is crucial to the final algorithm, but is presented with no proof or explanation.”
>
>
> Equation 8 is a well-known result with detailed proofs provided in [1], [2]. We felt it more appropriate to refer readers to these works rather than repeating the results in the appendix.
>
> -------------------------------------------------------------------------------------------------------------------------
>
> “Just above theorem 1 the sentence does not parse "Further, for each s, let λs be the solution to ", firstly there is no 'solution' to an equation, secondly should it be λs or pi?”
>
> Just above Theorem 1, note that 𝜋^𝝀 is defined as a function of 𝝀. We then define 𝝀_s as the  𝝀 that makes the value of the KL divergence equal to epsilon.
>
> -------------------------------------------------------------------------------------------------------------------------
>
> “The discussion following theorem 1 is very messy and hard to follow and the notation is horrendous. I'm confused as to why the indicator function in the 'disaggregated' update only includes states for which the constraint is already satisfied, what about the states where it is not? I presume this is because you initialize from the previous policy, but this seems very approximate and even worse updating the parameters for one state might significantly move the policy in some other state meaning large violations are possible and not dealt with.”
>
> Please note that the indicator function in Equation (18) is applied to each state in the trajectories sampled by the current policy. The indicator function helps us to take into account the disaggregated constraints (14) in the optimization problem (12)-(14).
>
> Please also note that the original policy improvement bound from TRPO [1] [2] were proved using the disaggregated constraints. We would like to quote the TRPO paper: “This problem imposes a constraint that the KL divergence is bounded at every point in the state space.” In the TRPO, the disaggregated constraint is too unwieldy to work with and they thus choose to work with the average KL divergence (the aggregated KL constraints in our case). In our paper, we work directly with the disaggregated constraint, but make the approximation that we only enforce the disaggregated constraints for the states in the trajectories sampled by the current policy.
>
> Thank you for your comment here! We will rewrite the discussion to better explain our reasoning regarding the indicator function.
>
> -------------------------------------------------------------------------------------------------------------------------
>
> “The connections to the papers 'MAXIMUM A POSTERIORI POLICY OPTIMISATION' and 'Relative Entropy Policy Search' should be mentioned, as another commenter said previously.”
>
> In the revision, we will include a discussion of MPO and Relative Entropy Policy Search and their relationship to our work.
>
> -------------------------------------------------------------------------------------------------------------------------
>
> “I don't think TRPO/PPO is SOTA anymore, so maybe these baselines aren't particularly interesting.”
>
> We acknowledge that recent work such as SAC has show to improve performance on TRPO and PPO, however we would like to point out that the main focus of our paper is to explore the idea of separating finding the optimal policy into a two-step process: finding the optimal non-parameterized policy, and then parameterizing this optimal policy. As such, we wanted to compare with algorithms operating under the same algorithmic constraints, one of which is being on-policy. It is a general trend in RL that the performance of an on-policy algorithm can be improved by incorporating off-policy training (such as in SAC), We thus leave the extension of our approach to off-policy training to future work. We also invite the community to join us in making this extension. To help with this effort, we will release code for push-button replication of the main results in the paper.
>
> Thank you again for your input, we look forward to further discussions.
>
>
> Reference
>
> [1] Joshua Achiam, David Held, Aviv Tamar, and Pieter Abbeel. Constrained policy optimization. In International Conference on Machine Learning, pp. 22–31, 2017.
>
> [2] John Schulman, Sergey Levine, Pieter Abbeel, Michael Jordan, and Philipp Moritz. Trust region policy optimization. In International Conference on Machine Learning, pp. 1889–1897, 2015.

---

### Official Review · AnonReviewer2 · 2018-11-01
**Strong similarities to previous work with no comparison**

**Rating:** 6
**Confidence:** 4

**Review:**

The authors formulate policy optimization as a two step iterative procedure: 1) solving a constrained optimization problem in the non-parameterized policy space, 2) using supervised regression to project this onto a parameterized policy. This approach generally applies to both continuous and discrete action spaces and can handle a variety of constraints. Their primary claims is that this approach improves sample-efficiency over TRPO on Mujoco tasks and over PPO on Atari games.

The method proposed in the paper has strong similarities with existing methods, but lacks comparisons with these approaches. The authors have not clearly demonstrated that SPU provides novel insights beyond the existing literature. I'm happy to change my score if the authors can convince me otherwise.

Main comments:

The focus of the paper is sample-efficiency, but the intro restricts to the on-policy setting. The authors should justify this choice. It is well known that off-policy algorithms (e.g., SAC for continuous control and Implicit Quantile Networks for Atari) are much more sample-efficient.

In Sec 4, what is the advantage of breaking the problem up into these 3 steps versus directly trying to solve (9),(10)? In fact, if we convert (10) into a penalty and take the derivative, we arrive at nearly the same gradient as (17). As this is central to the SPU framework, this needs to be justified.

MPO (Abdolmaleki et al. 2018) is very closely related to SPU. It is unclear if SPU provides any additional insights or benefits over MPO. This needs to be discussed and compared.

The experimental section could be strengthened by:
* Given the similarity to SPU, comparisons to MPO and GAC should be made, or clear justification for why they are not comparable must be given.
* Why is the comparison on Mujoco to TRPO in the main text and the comparison to PPO relegated to the appendix? It would make more sense to compare to PPO, so the authors need to justify this decision.
* The results on Mujoco are quite poor compared to state-of-the-art methods (e.g., SAC). The authors should justify why we should care about their results.

Comments:

In Sec 2, the authors should be careful about the discounting. For example, they are almost surely not having A_{it} approximate \gamma^t A^{\pi_{\theta_k}}, rather A^{\pi_{\theta_k}}.

In Sec 2, the KL is denoted as KL(\pi || pi_k), but in the text is described as the KL from \pi_k to \pi (reversed). From the equations, it appears that is an error, and it should read KL from \pi to \pi_k.

In Sec 3, the description of NPG/TRPO is not accurate. The main goal of NPG/TRPO work was to establish monotonic improvement.

In Sec 3, computational speed is cited as a major deficit of GAC, especially the solving linear systems with the Hessian (wrt to the actions). This seems rather surprising. Inverting a 1000x1000 matrix on a modern computer takes <1 second, so it doesn't seem like this should be the limiting step for any of the problems encountered.

The KL penalty version of PPO seems closely related to SPU. Can the authors mention differences with that version of PPO in the related work?

In Sec 4, step iii is described as supervised learning. Can the authors elaborate on why? I would typically think of the other direction as supervised learning as that leads to MLE.

In Sec 5.1, what is the justification/reasoning for setting \tilde{\lambda_{s_i}} = \lambda and introducing the indicator functions?

Sec 5.2 is not evaluated and Sec 5.3 produces inferior results, so it may make sense to move these to the appendix. Otherwise, the authors should explain situations where we would expect these to be useful or provide some additional insight. It also should be noted that the proximity constraints in TRPO/PPO follow from a theoretical argument and are not arbitrary choices.

Sec 5.3 seems to deviate quite a bit from the SPU framework. In addition to the differences pointed out in the text, the "supervised" loss changes too. Can the authors justify/explain the reasoning for these changes?

=====

I appreciate the authors' efforts to improve the paper. However, there is still substantial room for improvement in writing clarity. For example, the authors optimize the reverse KL from typical supervised learning, which makes even the title of the method confusing. The method that was experimentally evaluated can be derived more simply without the two-step procedure by directly taking the gradient and add the heuristically motivated per state indicator. This in itself is interesting and the authors demonstrate that it works well experimentally. I think the paper would be substantially more useful to the community if the authors focused on that contribution alone. As it stands now, I find the paper difficult to read because most of the theoretical results have no bearing on the method.

---

> ### Author Response · Authors · 2018-11-10
> **Reply to your comments (general reply)**
>
> Thank you for your thorough comments!
>
> We argue that SPU has the following advantages:
>
> -  It is mathematically principled.
>
> -  It is mathematically straightforward. It is much simpler and more understandable than MPO and GAC. The technique is comprehensible by undergraduates. Simplicity in RL algorithms has its own merits. This is especially useful when non-expert wants to apply RL algorithms to solve their problems, which is becoming a trend. The step-by-step description and implementation of SPU in only slightly more involved than PPO.
>
> -  It is versatile since a wide variety of constraint types can be used. SPU provides a general framework, allowing practitioners to try out different constraint types. For example, the TRPO paper mentions that enforcing the disaggregated constraint is preferable to enforcing the aggregated constraints. However, for mathematical conveniences, they choose to work with the aggregated constraints: “While it is motivated by the theory, this problem is impractical to solve due to the large number of constraints. Instead, we can use a heuristic approximation which considers the average KL divergence” [6]. In our paper, we show that the SPU framework allows us to solve the optimization problem with the dis-aggregated constraint exactly and also experimentally demonstrates that doing so helps to bring performance of SPU to be higher than TRPO.
>
> -  It is desirable to have a unified algorithm that applies to both continuous and discrete problems. In our understanding, GAC and SAC do not apply to the discrete case. MPO has only made preliminary progress with the discrete case.
>
> -  To our knowledge, among all the on-policy algorithms, it gives the best performance for the continuous cases. Deeply understanding the on-policy case is beneficial to the community. In future work we will consider combining SPU with replay buffers.
>
> -  It is a general approach to solving DRL problems. The algorithms tested in our paper are specific instances of this approach. In both the GAC and MPO paper, working in the non-parameterized policy space is a by-product of applying the main ideas in those papers to DRL.
>
> We will cite and discuss SAC and MPO paper in related work. We also argue that SAC is not directly comparable to SPU since SAC tunes environment-specific parameter (reward scaling). In SPU, the hyper-parameters are shared and fixed across all environments. The neural networks used in the SAC paper are also 4 times larger than the neural networks used in our work and the versions of PPO and TRPO that we compared against. Also, the performance of SPU and SAC can not be compared by looking at the graphs in the respective papers since the Mujoco environments used in the SAC paper is version 1 while we used version 2.
>
> That being said, in our paper, we want to compare SPU against algorithms that operate under the same constraints, one of which is being on-policy. Thus, the focus of the paper will remain comparing SPU with other on-policy schemes. We justify why in more detailed below.

---

> > ### Author Response · Authors · 2018-11-10
> > **On the advantage of working in the non-parameterized policy space**
> >
> > “In Sec 4, what is the advantage of breaking the problem up into these 3 steps versus directly trying to solve (9), (10)? In fact, if we convert (10) into a penalty and take the derivative, we arrive at nearly the same gradient as (17). As this is central to the SPU framework, this needs to be justified.”
> >
> > Response: A central aspect of SPU is its versatility, including to be able to handle both aggregated and disaggregated constraints. The three-step procedure allows for this diversity. Also, obtaining the optimal solution for the non-parameterized problems sheds significant insight on the problem. Also, the form of the gradient being nearly the same is not the same. If we convert (10) into a penalty and take the derivative, it is true we arrive at nearly the same gradient as (17). This is in fact one of the approach tried in the PPO paper, which they demonstrate does not work as well as the clipping approach in PPO. However, we show that in SPU, the form of gradient in (17) is superior to PPO.

---

> > > ### Comment · AnonReviewer2 · 2018-11-20
> > > **Re: On the advantage of working in the non-parameterized policy space**
> > >
> > > The key difference seems to be the disaggregated constraints, but these are added by per-state acceptance identity function, which as far as I can tell is an ad-hoc addition to account for the disaggregated constraints, unrelated to the two-step approach. Other than that, Equation 18 is exactly what you get from taking the derivative of the penalty version of the objective. If that is not correct, can the authors clarify?
> > >
> > > I agree that the authors do demonstrate that the disaggregated constraints are beneficial.

---

> > > > ### Author Response · Authors · 2018-11-20
> > > > **Response**
> > > >
> > > > We agree that Equation 18, minus the indicator variable, is what comes out if we take derivative of the penalty version of the objective. We argue that this does not represent a lack of novelty because:
> > > >
> > > > -  PPO considers a similar form of gradient, with adaptive KL penalty. However, in PPO, this form of gradient is inferior to PPO clipping. To the best of our knowledge, it is only until our work that such form of gradient is demonstrated to outperform PPO clipping.
> > > >
> > > > -  The two-step procedure allows us to theoretically motivate adding the indicator variable. Considering what would happen if we were to add the indicator variable to either TRPO or PPO without the two-step process and the optimization problem (12-14). In TRPO, theta_k is only updated once to obtain theta_{k+1}. Thus, it does not make sense to add the indicator. In PPO, the disaggregated constraint is not considered. In our paper, the indicator variable is intimately linked to the two-step procedure and the optimization problem (12-14). The two-step procedure allows us to solve this problem exactly. To have the gradient to update the policy, we make approximations. The indicator variable helps to enforce hard constraints despite the approximations.

---

> > > > > ### Comment · AnonReviewer2 · 2018-11-20
> > > > > **Re: Response**
> > > > >
> > > > > Sorry, I'm not understanding something. Maybe I can explain my understanding and you can explain where it is wrong.
> > > > >
> > > > > 1. Equations 12, 13 are the KL PPO objective. Equation 14 is a novel constraint introduced by the authors. At this point, the problem has nothing to do with the two-step procedure.
> > > > > 2. For solving the problem with Equations 12 & 13, we can convert the constraint to a penalty and take the derivative. This gives Eq. 18 without the indicator.
> > > > > 3. The authors introduce the indicator as a way to enforce the constraint in Eq. 14. This is heuristic and while it makes sense intuitively is not justified from a theoretical perspective.
> > > > > 4. Alternatively, the two-step procedure can be used and we again have to make approximations and we end up at Eq. 18. Is the indicator approximation motivated by the optimal solution (Eq. 15)? This is not mentioned in the text.
> > > > >
> > > > > It seems that Eq. 18 can be motivated by 1, 2, 3 without ever mentioning the two-step procedure. I agree that the authors show that the new constraint (Eq. 14) is helpful, but then the paper should be about that constraint and not about a two-step procedure. This is what I'm unclear on.

---

> > > > > > ### Author Response · Authors · 2018-11-20
> > > > > > **Response**
> > > > > >
> > > > > > We applaud the reviewer for investing the time to understand our paper.
> > > > > >
> > > > > > We *do not* introduce Equation 14. Equation 14, the disaggregated constraint, is the constraint that TRPO wants to but is unable to enforce (Equation 11 in TRPO paper). To be able to solve the optimization problem, TRPO replaces the disaggregated constraint with the aggregated constraint (Equation 12 in TRPO paper). As such, Equation 14 is *more* theoretically justified than Equation 13.
> > > > > >
> > > > > > Disadvantaged because they work in the parameter space, TRPO also needs to make 2 other approximations (linear approximation to objective and quadratic approximation to aggregated constraints). The two-step procedure allows us to bring in convex theory and duality to solve exactly the original problem TRPO wants to but is unable to solve. To our knowledge, this would not have been possible without the two-step procedure.
> > > > > >
> > > > > > Thus, compared to PPO, we are on significantly stronger theoretical footing. This is also reflected in our superior performance compared to PPO.
> > > > > >
> > > > > > We need to make approximation to obtain the stochastic gradient, as is any stochastic gradient descent algorithms. The indicator helps to enforce hard constraints despite the approximations. However, we agree that we do not have formal proof on how the indicator variable affects the policy performance bound. This is why we ran extensive ablation studies and experimental comparison against TRPO and PPO.

---

> > > > > > > ### Comment · AnonReviewer2 · 2018-11-22
> > > > > > > **Re: Response**
> > > > > > >
> > > > > > > Does the non-parameterized solution explain the use of the indicator function approximation? Otherwise, while it's nice to have a theoretical solution to the problem in terms of intractable quantities, getting to Eq. 18 can equally be done by starting from 12-14, adding the penalty, taking the derivative, and adding the indicator approximation. Is there a practical benefit to having the theoretical solution?

---

> > > > > > > > ### Author Response · Authors · 2018-11-22
> > > > > > > > **Response**
> > > > > > > >
> > > > > > > > The non-parameterized solution does not directly explain the use of the indicator function. However, the methodology provides theoretical understanding and a concrete framework for further analysis, which we argue is an important scientific contribution.
> > > > > > > >
> > > > > > > > Having good theory makes it easier for the community to analyze and build upon scientific works. For example, the original problem in TRPO is intractable and the TRPO algorithm uses several approximations.  However, the theoretical analysis in TRPO opens the door for further works (PPO, CPO, etc) and therefore has high impact (cited >750 since 2015).
> > > > > > > >
> > > > > > > > The other benefit we demonstrated is the advantage of the two-step procedure. We take a difficult optimization problem over the parameter space, and show how it can be solved by converting to an optimization problem over the output space. This is a general technique. By comparing with TRPO, both in theory and in experiment, we demonstrated its benefit.
> > > > > > > >
> > > > > > > > Although we consider three classes of proximity constraint, there may be yet another class that leads to even better performance. The methodology allows researchers to explore other proximity constraints in the future.

---

> > ### Author Response · Authors · 2018-11-10
> > **On MPO**
> >
> > “MPO (Abdolmaleki et al. 2018) is very closely related to SPU. It is unclear if SPU provides any additional insights or benefits over MPO. This needs to be discussed and compared.”
> >
> > Response: MPO uses much more sophisticated machinery, namely, Expectation Maximization to address the DRL problem. Our approach is more straightforward, and has been designed to handle different types of trust region constraints in a natural manner. Also, the approach naturally applies to discrete (Atari) as well as continuous (Mujoco). Also, we argue that working in the non-parameterized space is a general framework to solving DRL problem and demonstrates its theoretical and experimental benefits. In MPO, working in the non-parameterized space is a by-product of applying EM to DRL problems.

---

> > > ### Comment · AnonReviewer2 · 2018-11-19
> > > **Re: On MPO**
> > >
> > > The fact that MPO is motivated by a high level idea and the algorithm naturally follows from it is a positive.

---

> > ### Author Response · Authors · 2018-11-10
> > **Comparison against SAC and state-of-the-art approach**
> >
> > “Given the similarity to SPU, comparisons to MPO and GAC should be made, or clear justification for why they are not comparable must be given.
> > * Why is the comparison on Mujoco to TRPO in the main text and the comparison to PPO relegated to the appendix? It would make more sense to compare to PPO, so the authors need to justify this decision.
> > * The results on Mujoco are quite poor compared to state-of-the-art methods (e.g., SAC). The authors should justify why we should care about their results.”
> >
> > Response: We do not claim state of the art results. Rather, we hope our paper ignites interests in separating finding the optimal policy into a two-step process: finding the optimal non-parameterized policy, and then parameterizing this optimal policy. As such, we wanted to compare with algorithms operating under the same algorithmic constraints, one of which is being on-policy. Also, both SAC and Implicit Quantile Networks for Distributional Reinforcement Learning have a long of history of development and we argue this is why they are state-of-the-arts method. In contrast, SPU is among the first few papers to consider working in the non-parameterized space.
> >
> > It is a general trend in RL that the performance of an on-policy algorithm can be substantially improved by incorporating off-policy training. For example, experience replay was used to make Q-learning the state-of-the-art approach in DRL [1]. Subsequently, experience replay was also incorporate into actor-critic methods, which brought its performance to being equal to Q-learning method [2]. We thus leave the extension of our approach to off-policy training to future work. We also invite the community to join us in making this extension. To help with this effort, we will release code for push-button replication of the main results in the paper.
> >
> > We included the graphs for Atari in the appendix because there are too many of them. We have included figure 2 to provide for a high-level overview of the performance of SPU vs PPO. We will rewrite the explanation of figure 2 to make this clearer.
> >
> > We acknowledge that solving the model-free DRL problem as a two-step procedure of first solving a non-parameterized problem and then a projection onto the parameter space has independently been proposed in the GAC paper and the MPO paper. We will be sure to give the MPO paper proper credit in the revision.
> >
> > We acknowledge SAC and other off-policy techniques can do better than SPU, which is on policy. We thank the reviewer for pointing out to us the SAC paper. We will make sure to cite SAC and discuss its benefit in our related work section. However, we argue that by solving the same problem that TRPO tries to solve and out-performing it with the only change being working in the non-parameterized space, we have demonstrates that our approach leads to experimental gain in a controlled experimental setting.

---

> > > ### Comment · AnonReviewer2 · 2018-11-19
> > > **Re: Comparison against SAC and state-of-the-art approach**
> > >
> > > But, why are you introducing this constraint? There needs to be a clear motivation for why this two-step procedure would be a good thing to do. You motivate the work by sample-efficiency concerns, but the proposed approach is far from the state-of-the-art in sample-efficiency. A reader of your paper will be confused as to why they should care about your results. Reworking the motivation to justify your approach will greatly strengthen the paper.
> > >
> > > Your approach is more similar to PPO than TRPO. So, the Mujoco PPO results should be in the main-text, not the Mujoco TRPO results.

---

> > > > ### Author Response · Authors · 2018-11-20
> > > > **Response**
> > > >
> > > > We will include Mujoco PPO result in the main text.
> > > >
> > > > The motivation for this two-step procedure is that it allows us to solve the optimization problem exactly. Our experimental result demonstrates the benefit. We compared against TRPO in a *controlled* experimental setting, including trying to solve the same optimization problem, same network size, same advantage estimation scheme, etc. Since we clearly outperform TRPO, we argue that this two-process procedure has significant potentials. To put this in the larger context of our field, constrained policy optimization is becoming a trend [7, 8, 9]. It would be nice, for both theoretical understanding and empirical performance, to have a procedure that allows us to solve the optimization problem exactly.

---

> > ### Author Response · Authors · 2018-11-10
> > **On the computational speed of GAC**
> >
> > “In Sec 3, computational speed is cited as a major deficit of GAC, especially the solving linear systems with the Hessian (wrt to the actions). This seems rather surprising. Inverting a 1000x1000 matrix on a modern computer takes <1 second, so it doesn't seem like this should be the limiting step for any of the problems encountered.”
> >
> > We profiled GAC and identified these major computation bottlenecks. At every policy and q-network update step, GAC:
> >
> > -  Forms empirical estimate of the Q-value by sampling multiple actions from the action distribution and running a forward pass for each action (30% of total computational time) [3]
> >
> > -  Minimizes the dual function to obtain the dual variables using SLSQP (20% of total computational time) [4]
> >
> > -  Forms taylor approximation of the Hessian (10% of computational time) [5]

---

> > ### Author Response · Authors · 2018-11-10
> > **References used in our reply**
> >
> > [1] Human-level control through deep reinforcement learning. https://storage.googleapis.com/deepmind-media/dqn/DQNNaturePaper.pdf
> >
> > [2] Sample Efficient Actor-Critic with Experience Replay https://arxiv.org/pdf/1611.01224.pdf
> >
> > [3] https://github.com/voot-t/guide-actor-critic/blob/8af08348a83f5f0cf32f70347fd987fb4391e53d/GAC_learner.py#L166
> >
> > [4] https://github.com/voot-t/guide-actor-critic/blob/8af08348a83f5f0cf32f70347fd987fb4391e53d/GAC_learner.py#L245
> >
> > [5] https://github.com/voot-t/guide-actor-critic/blob/8af08348a83f5f0cf32f70347fd987fb4391e53d/GAC_learner.py#L231
> >
> > [6] TRPO. https://arxiv.org/pdf/1502.05477.pdf
> >
> > [7] Joshua Achiam, David Held, Aviv Tamar, Pieter Abbeel. Constrained Policy Optimization.
> >
> > [8] Voot Tangkaratt, Abbas Abdolmaleki, Masashi Sugiyama. Guide Actor-Critic for Continuous Control.
> >
> > [9] Chen Tessler, Daniel J. Mankowitz, Shie Mannor. Reward Constrained Policy Optimization.

---

> > ### Comment · AnonReviewer2 · 2018-11-19
> > **Re:**
> >
> > Is the claim that it is more principled than other approaches?
> >
> > "The technique is comprehensible by undergraduates. Simplicity in RL algorithms has its own merits. This is especially useful when non-expert wants to apply RL algorithms to solve their problems, which is becoming a trend." and "deeply understanding the on-policy case is beneficial to the community." are both reasonable motivations for your work. I hope you can rework the introduction to motivate the work by that reasoning instead of the current version.
> >
> > It's unclear why SAC wouldn't apply to the discrete case. However, it would be quite similar to MPO at that point.  Yes, I agree that directly comparing to the SAC paper graphs is not possible, but multiple open source versions of SAC exist, including from the authors of the original paper.
> >
> > The fact that GAC and MPO derive the non-parameterized policy space as a by-product of applying higher level ideas is a positive.

---

> > > ### Author Response · Authors · 2018-11-20
> > > **Response**
> > >
> > > We will rework the introduction to include those motivations. Regarding SAC, we reiterate that for this paper, we want to compare against algorithms operating under the same algorithmic constraints, one of which is on-policy. It would actually not be surprising if SAC outperforms SPU since SAC uses past samples and SPU does not. But we argue that such comparison does not shed additional insight into our work because we are not claiming state-of-the-arts result.

---

> ### Author Response · Authors · 2018-11-29
> **Post-rebuttal question to reviewer**
>
> Dear reviewer,
>
> We thank you for the discussion. We have made your suggested changes to the paper, including reworking the motivations and adding PPO Mujoco result in the experimental section.
>
> We were wondering if you can change your review score to reflect the changes we have made. Thanks!

---

### Official Review · AnonReviewer3 · 2018-11-06
**A very interesting approach to constraint policy optimization that uses nonparametric relaxation with subsequent projection**

**Rating:** 9
**Confidence:** 2

**Review:**

The paper proposes to perform a constraint optimization of an approximation of the expected reward function for unparameterized policy with subsequent projection of the solution to the nearest parameterized one. This approach allows fast ("nearly closed form") solutions for nonparametric policies and leads to an increase in sample efficiency.

The proposed approach is interesting and the results are promising.

---

> ### Author Response · Authors · 2018-11-10
> **Reply to your comments**
>
> Thank you so much for your comments! We are more than happy to answer any thing about the paper should you have more comments in the future.

---

### Public Comment · (anonymous) · 2018-09-30
**Supervised policy update**

Hello,

It seems there are potential algorithmic similarities with a paper from  ICLR2018:

Maximum a-Posteriori Policy Optimisaiton
https://arxiv.org/pdf/1806.06920.pdf

Here  the policy optimization has two steps:

1- E-Step: Computing non-parametric policies (See equations 7,8 and 9 of the paper)
2- M-step: Using supervised learning to fit a parametric policy to non-parametric ones from E-step (see equation 12).

Please also see "Relative Entropy Policy Search (AAAI2009)":

https://www.aaai.org/ocs/index.php/AAAI/AAAI10/paper/viewFile/1851/2264

Thank you,

---

> ### Author Response · Authors · 2018-10-04
> **Answer**
>
> Hi,
>
> Thank you for your comment! Our paper does indeed involve first finding a non-parameterized policy, then using supervised learning to fit a parameterized policy to the nonparametric one, similar to what you described in the maximum a posteriori optimization paper. However, the way with which we derived the non-parametric and parametric policies are fundamentally different, instead of an inference-based approach, our algorithms were derived from a proximal optimization perspective. Our approach can address a wide-variety of different definitions of proximity, including aggregated and disaggregated KL proximity, reverse KL proximity, L-infinity proximity. Each definition of proximity leads to a different non-parameterized optimal policy.

---

> > ### Public Comment · (anonymous) · 2018-10-25
> > **Re: Answer**
> >
> > Thanks for the reply.
> >
> > 2-step optimization process in MPO [1] has been derived from a single inference principle. And the derivation naturally gives guidance on which direction of KL should be used in E-step which accounts for deriving the non-parametric policies. I think given the tight connections between MPO [1], REPS [2]  and your method in terms of update rules, a discussion on similarities and differences would be informative.
> >
> > Thank you,
> >
> > [1] https://arxiv.org/pdf/1806.06920.pdf
> > [2] https://www.aaai.org/ocs/index.php/AAAI/AAAI10/paper/viewFile/1851/2264

---

### Public Comment · (anonymous) · 2018-10-03
**Derivation from Eq.17 to Eq.18**

Hi,
Thank you for the great paper and the detailed explanation. This is so far the best ICLR paper that I read. However, I didn't quite follow the simplification from Eq. 17 to Eq. 18. In Eq.7, you would penalize the advantage term when KL is violated. But in Eq.18, you just zero out the gradient when the KL is violated?

---

> ### Author Response · Authors · 2018-10-03
> **Clarifications**
>
> Hi,
>
> We're happy you enjoyed reading our paper. To ensure we answer your questions correctly, we would like to clarify the following: in the sentence "In Eq.7, you would penalize the advantage term when KL is violated.", are you referring to equation 17 instead of equation 7?

---

> > ### Public Comment · (anonymous) · 2018-10-03
> > **Clarifications on Clarifications**
> >
> > Sorry my bad. I meant "In Eq.17, you would penalize the advantage term when KL is violated."

---

> > > ### Author Response · Authors · 2018-10-04
> > > **Answer**
> > >
> > > In eq 17, penalizing the advantage term comes from the derivation of the gradient of the KL (as shown in appendix B1). In eq 18, for each sampled state s_i, the contribution of the gradient coming from s_i is zeroed out if the KL constraint for s_i is violated. Hope that answers your question!

---

### Author Response · Authors · 2018-11-26
**Summary of changes**

We would like to thank the reviewers for their feedback. We have made the changes you requested. This post summarises the major changes.

Section 1 (Introduction):

-   Emphasize that our work aims to strike the right balance between simplicity and performance.

-   Discuss off-policy algorithms such as SAC.

-   Emphasize that we want to focus on comparing against on-policy algorithms in this paper and that we do not claim state-of-the-art sample efficiency result.

Section 3 (Related work):

-   Mention MPO, PPO with KL penalty and Relative Policy Entropy Search and discuss their similarities and differences compared to our paper.

Section 5 (SPU applied to specific proximity criteria):

-   Discuss the motivation of the optimization problem (12)-(14) from TRPO’s perspectives.

-   Highlight how the two-step procedure allows us to solve the optimization problem exactly, thereby offering theoretical understanding and a framework for constrained policy optimization.

-   Theorem 1 and surrounding text are rewritten for clarity.

Section 6 (Experimental results):

-   PPO Mujoco result is included in the comparison with our approach.

-   The description of Figure 2 is rewritten to help with interpreting the figure.

Appendix B:

-   Mathematical derivations are followed by explanations of what were used to get the next steps.

---

### Meta-Review · Area_Chair1 · 2018-12-13
**Accept**

**Confidence:** 4
**Recommendation:** Accept (Poster)

**Metareview:**

The paper presents an interesting technique for constrained policy optimization, which is applicable to existing RL algorithms such as TRPO and PPO. All of the reviewers agree that the paper is above the bar and the authors have improved the exposition during the review process. I encourage the authors to address all of the comments in the final version.